# Minimal presynaptic protein machinery governing diverse kinetics of calcium-evoked neurotransmitter release

Dipayan Bose[1,2,6], Manindra Bera [1,3,6], Christopher A. Norman [4,5], Yulia Timofeeva [5], Kirill E. Volynski [3,4] ✉ & Shyam S. Krishnakumar [1,2,4] ✉

Neurotransmitters are released from synaptic vesicles with remarkable precision in response to presynaptic calcium influx but exhibit significant heterogeneity in exocytosis timing and efficacy based on the recent history of activity. This heterogeneity is critical for information transfer in the brain, yet its molecular basis remains poorly understood. Here, we employ a biochemically-defined fusion assay under physiologically relevant conditions to delineate the minimal protein machinery sufficient to account for various modes of calcium-triggered vesicle fusion dynamics. We find that Synaptotagmin-1, Synaptotagmin-7, and Complexin synergistically restrain SNARE complex assembly, thus preserving vesicles in a stably docked state at rest. Upon calcium activation, Synaptotagmin-1 induces rapid vesicle fusion, while Synaptotagmin-7 mediates delayed fusion. Competitive binding of Synaptotagmin-1 and Synaptotagmin-7 to the same SNAREs, coupled with differential rates of calcium-triggered fusion clamp reversal, govern the overall kinetics of vesicular fusion. Under conditions mimicking sustained neuronal activity, the Synaptotagmin-7 fusion clamp is destabilized by the elevated basal calcium concentration, thereby enhancing the synchronous component of fusion. These findings provide a direct demonstration that a small set of proteins is sufficient to account for how nerve terminals adapt and regulate the calcium-evoked neurotransmitter exocytosis process to support their specialized functions in the nervous system.

Information transfer in the brain depends on the release of neurotransmitters stored in synaptic vesicles (SVs) within the presynaptic terminals. SV fusion with the presynaptic membrane and neurotransmitter release are tightly regulated by changes in the presynaptic $Ca^{2+}$ concentration ($[Ca^{2+}]$) and can occur in less than a millisecond after the action potential (AP) invades a presynaptic terminal[1,2]. In addition to fast, synchronous release that keeps pace with APs, many synapses also exhibit delayed asynchronous release that persists for tens to hundreds of milliseconds[1,2]. Synapses also vary in terms of how the probability of neurotransmitter release is altered by the recent history of AP firing[3,4]. The balance between synchronous and asynchronous release, and the degree of synaptic facilitation or depression of release, differs not only among neurons but also among synapses supplied by a single axon according to their postsynaptic targets. This heterogeneity is important in coordinating activity within neuronal networks[5–7].

[1]Nanobiology Institute, Yale University, West Haven, CT, USA. [2]Department of Neurology, School of Medicine, Yale University, New Haven, CT, USA. [3]Department of Cell Biology, School of Medicine, Yale University, New Haven, CT, USA. [4]Department of Clinical and Experimental Epilepsy, UCL Queen Square Institute of Neurology, London, UK. [5]Department of Computer Science, University of Warwick, Coventry, UK. [6]These authors contributed equally: Dipayan Bose, Manindra Bera. ✉e-mail: k.volynski@ucl.ac.uk; shyam.krishnakumar@yale.edu

The key components of the synaptic vesicular exocytosis machinery have been identified[1,2,8]. These include the SNARE proteins that catalyze SV fusion (VAMP2 on the SV, and Syntaxin/SNAP25 on the presynaptic membrane); Ca$^{2+}$ release sensors that couple SV fusion to Ca$^{2+}$ signal (Synaptotagmins); and proteins that regulate SV docking and the organization of vesicular release sites (e.g., Complexin (CPX), Munc13, Munc18). Ca$^{2+}$-evoked neurotransmitter release occurs from a readily releasable pool (RRP) of vesicles docked at the presynaptic active zone[1,2,9]. A consensus has emerged that, at an individual RRP vesicle, multiple SNARE complexes are arrested ('clamped') in a partially-assembled state (SNAREpins) by Synaptotagmins and CPX. Ca$^{2+}$ activation of Synaptotagmins releases the fusion clamp allowing SNAREpins to fully assemble and drive SV fusion[10–12]. Despite this general scheme, the reasons for variations in the synchrony of exocytosis or the occurrence of short-term facilitation or depression among synapses, remain poorly understood.

Synchronous neurotransmitter release, occurring within a few milliseconds of the arrival of an AP, is triggered by a transient high local [Ca$^{2+}$] ([Ca$^{2+}$]$_{peak}$ ~ 10-100 μM) at vesicular release sites, and genetic deletion and substitution experiments have shown that it requires a fast, low-affinity Ca$^{2+}$ sensor such as Synaptotagmin 1, 2 or 9 (Syt1, Syt2, Syt9)[13,14]. Asynchronous neurotransmitter release can occur in response to a single AP but is particularly prominent during and following high-frequency bursts of APs. This delayed release requires a persistent elevation of presynaptic [Ca$^{2+}$] and this [Ca$^{2+}$]$_{residual}$ is thought to reach a low micromolar concentration[1,15,16]. At many synapses, accumulation of [Ca$^{2+}$]$_{residual}$ also leads to transient facilitation of the fast synchronous release component[1,4]. The slow, high-affinity Ca$^{2+}$ sensor Synaptotagmin 7 (Syt7), which is activated by both [Ca$^{2+}$]$_{peak}$ and [Ca$^{2+}$]$_{residual}$, has been implicated in regulating both asynchronous release and short-term facilitation[17–20]. Indeed, previous studies have shown that genetic removal of Syt7 reduces short-term synaptic facilitation and asynchronous release[13,18,19]. However, the Syt7's role in asynchronous release remains a topic of debate as other Ca$^{2+}$-release sensors (e.g. Syt1, Syt3 and Doc2A) have also been implicated in mediating asynchronous release component[17,21–23].

An inherent limitation of genetic studies is their inability to demonstrate whether Syt1 and Syt7 alone are *sufficient* to regulate the timing and plasticity of neurotransmitter release, as the contribution of other presynaptic proteins cannot be ruled out. Furthermore, because vesicular exocytosis involves an interplay of presynaptic Ca$^{2+}$ dynamics and Ca$^{2+}$ sensors, a quantitative account of synchronous/asynchronous release kinetics and short-term plasticity requires precise control and measurement of [Ca$^{2+}$]. This is difficult to achieve in intact synapses owing to the small size of the active zone and the high speed of Ca$^{2+}$ diffusion and buffering[24,25].

Hence, we sought to determine whether Syt1 and Syt7 (along with SNAREs and CPX) are sufficient to determine the kinetics and activity-dependent changes in Ca$^{2+}$-evoked SV release. Additionally, we aimed to uncover the underlying molecular mechanisms governing the cooperative action of Syt1 and Syt7. To achieve this, we took a reductionistic approach of combining an in vitro reconstituted fusion assay[26–29] with quantitative computational modeling[12]. Specifically, we utilized a biochemically-defined high-throughput assay based on a suspended lipid membrane platform that uses fluorescence microscopy to track the docking, clamping (equivalent to the delay from docking to spontaneous fusion), and Ca$^{2+}$-triggered fusion of individual vesicles at tens of milliseconds precision. Critically, this setup allowed precise control over the identity and density of the included proteins, as well as the [Ca$^{2+}$] signal[26,28–30].

We report that under resting conditions, Syt1 and Syt7 along with CPX clamp vesicle fusion to produce docked vesicles. Upon Ca$^{2+}$-influx, Syt1 and Syt7 act as 'fast' and 'slow' release sensors respectively and govern the overall fusion kinetics by competitively binding to the same SNARE complex. Computational modeling suggests that the slower Ca$^{2+}$-triggered reversal of the Syt7 fusion clamp, in contrast to Syt1, accounts for the delayed fusion kinetics. When [Ca$^{2+}$]$_{basal}$ is elevated to mimic neuronal activity, Syt7 enhances Ca$^{2+}$-synchronized vesicle fusion independent of Syt1, due to selective destabilization of the Syt7 clamp by elevated [Ca$^{2+}$]$_{basal}$. In summary, our data suggest that Syt1, Syt7, SNAREs, and CPX constitute the minimal protein machinery necessary to support the diverse kinetics and activity-dependent dynamics of Ca$^{2+}$-evoked neurotransmitter release.

## Results

Recently, using the in vitro experimental setup, we demonstrated that under physiologically relevant conditions, Syt1 and CPX are sufficient to produce clamped (RRP-like) vesicles, and these stably docked vesicles can be triggered to fuse rapidly by Ca$^{2+}$ addition[30]. Building on this advance, we designed the reconstitution conditions to investigate the cooperative action of Syt1 and Syt7 as follows: in all experiments, we used small unilamellar vesicles containing VAMP2 and Syt1 and included CPX in the solution (Fig. 1a, Supplementary Fig. 1). We reconstituted pre-formed t-SNAREs (a 1:1 complex of Syntaxin1 and SNAP-25) and Syt7 (when warranted) in the suspended lipid membrane (Fig. 1a, Supplementary Fig. 1). We incorporated Syt7 in the suspended lipid membrane reflecting its predominant localization in the pre-synaptic membrane in central synapses[31,32]. Since the concentration of Syt7 within the active zone is unknown and likely varies among different types of synapses, we tested the effect of varying Syt7 concentration. In all cases, we monitored large ensembles of vesicles (~150 – 200 vesicles) and used fluorescently labeled lipid (2% ATTO647N-PE), introduced in the vesicles to track the docking and fate of individual vesicles (Fig. 1a and Methods). To trigger the fusion of docked vesicles, we chose a [Ca$^{2+}$] of 100 μM. This concentration aligns with the [Ca$^{2+}$]$_{peak}$ observed at presynaptic vesicular release sites[24] and is sufficient to saturate both Syt1 and Syt7[10], therefore mitigating possible variability stemming from differential activation of Syt1 and Syt7.

### Syt7 delays the fusion of Syt1-containing vesicles in a concentration-dependent manner

In the absence of Syt7, the majority (~95%) of the Syt1/VAMP2 containing vesicles that docked to the t-SNARE bilayers were 'immobile' and remained unfused during an initial 10 min observation window (Supplementary Fig. 2). Addition of Ca$^{2+}$ triggered the fusion of ~90% of the stably clamped vesicles within 5 s as measured by lipid mixing (Fig. 1b). Notably, a significant portion of fusion events (~70%) occurred within 2 frames (~300 ms) following the initial arrival of Ca$^{2+}$ signal (Fig. 1b, c), even though the [Ca$^{2+}$] reached 100 μM over time scale of ~750 milliseconds (Supplementary Fig. 3).

Inclusion of Syt7 in the t-SNARE-containing bilayer (at concentrations ranging from 1:2000 to 1:200 protein-to-lipid ratio) had no discernable effect on the number or the fate of the docked Syt1/VAMP2 vesicles (Supplementary Fig. 2). Hence, the vast majority (~90%) of the vesicles remained stably docked in an immobile clamped state (Supplementary Fig. 2). Likewise, Syt7 did not impact the Ca$^{2+}$-induced fusion competence as ~90% of the docked vesicles fused within 5 s following the addition of Ca$^{2+}$ (100 μM) (Fig. 1b). However, we observed significant delays in the kinetics of Ca$^{2+}$-triggered fusion, and these delays correlated with the amount of Syt7 included in the bilayer (Fig. 1c). The proportion of 'coupled release' i.e. vesicles undergoing fusion within 2 frames (~300 ms) following the initial arrival of Ca$^{2+}$ signal progressively declined from approximately 70% to 10%, as the concentration of Syt7 in the bilayer was increased from 1:2000 to 1:200 (Fig. 1b, c). Noteworthy, this impact was specific to Syt7, as inclusion of Syt1 (instead of Syt7) in the suspended bilayer, even at a protein-to-lipid ratio of 1:200, did not alter the likelihood or kinetics of Ca$^{2+}$-triggered fusion of Syt1/VAMP2 vesicles (Fig. 1b, c). Taken together, these data indicate that Syt7 influences the kinetics of Ca$^{2+}$-triggered

fusion for Syt1/VAMP2 vesicles in a concentration-dependent manner, without altering the fusion competence of docked vesicles.

## Contribution of Syt7 to the establishment of the fusion clamp

The clamping efficiency of docked vesicles was similar in the absence or presence of Syt7, with approximately 90% of docked vesicles stably clamped under both conditions (Supplementary Fig. 2). Therefore, it remained unclear whether Syt7 also contributes to the fusion clamp (in addition to Syt1 and CPX) under these conditions. Simple removal of Syt1 and/or CPX from the reaction mixture was not feasible, as omitting CPX potentiated spontaneous fusion, while leaving out Syt1 significantly reduced the number of docked vesicles, precluding any meaningful analysis[26,30]. Hence, we developed reconstitution conditions specifically tailored to investigate the role of Syt7 as a fusion clamp.

In previous work, we demonstrated that CPX could be omitted under low VAMP2 copy number conditions (i.e. vesicles containing ~13 copies of VAMP2 and ~22 copies of Syt1), as Syt1 alone could produce stably clamped $Ca^{2+}$-sensitive vesicles[27,30]. We further showed that disrupting the Syt1-SNARE interaction at the 'primary' interface using the well-established mutations in the Syt1 C2B domain (R281A, E295A, Y338W, R398A, R399A; referred to as Syt1^Q)[33,34] specifically abrogates the Syt1 fusion clamp without affecting vesicle docking[30]. Given this background, we investigated Syt7's impact on the fusion clamp by utilizing the Syt1^Q mutant in the CPX-free, low VAMP2 condition (i.e. Syt1^Q/VAMP2^low vesicles).

As anticipated, in the absence of Syt7, the majority (>90%) of docked Syt1^Q/VAMP2^low vesicles fused spontaneously. Inclusion of Syt7 in the bilayer restored the clamp on Syt1^Q/VAMP2^low vesicles in a dose-

dependent manner, with approximately 40% and 90% of vesicles remaining stably docked in an immobile state with Syt7 included at 1:800 and 1:200 (protein-to-lipid ratio) respectively (Fig. 2a). Furthermore, these stably docked vesicles could be triggered to fuse by the addition of 100 μM $Ca^{2+}$ (Fig. 2b), but the fusion kinetics were desynchronized from the $Ca^{2+}$ signal, with a temporally distributed vesicle fusion pattern (Fig. 2c). This suggests that Syt7 can independently establish a calcium-sensitive fusion clamp and may act in conjunction with Syt1 and CPX under physiologically relevant conditions.

## Syt1 and Syt7 regulate $Ca^{2+}$-evoked fusion via competitive binding to the SNARE complex

Next, we investigated the impact of $Ca^{2+}$-binding-deficient Syt1 and Syt7 mutants to understand the mechanisms behind their synergistic action (Fig. 3). Specifically, we employed Syt1 with D309A, D363A, D365A mutations in the C2B domain (Syt1^DA), and Syt7 with D225A, D227A, D233A, D357A, D359A mutations in the C2AB domains (Syt7^DA). The introduction of $Ca^{2+}$-insensitive Syt7^DA in the bilayer resulted in a concentration-dependent reduction in $Ca^{2+}$-evoked fusion of docked vesicles containing Syt1^WT/VAMP2, with ~30% decrease at a low (1:800 protein-to-lipid) and ~70% reduction at high (1:200) Syt7^DA concentrations (Fig. 3a). However, Syt7^DA had no discernable effect on the fusion kinetics, with the majority of vesicles fusing within the first 2 frames following $Ca^{2+}$ arrival (Fig. 3a).

As expected, the disruption of $Ca^{2+}$ binding to Syt1 (Syt1^DA) eliminated $Ca^{2+}$-triggered vesicular fusion (~7%) without altering the docking or clamping of the vesicles. However, the inclusion of Syt7^WT into the bilayer restored $Ca^{2+}$-evoked fusion to levels corresponding with the concentration of Syt7^WT in the bilayer (~40% and ~75% with

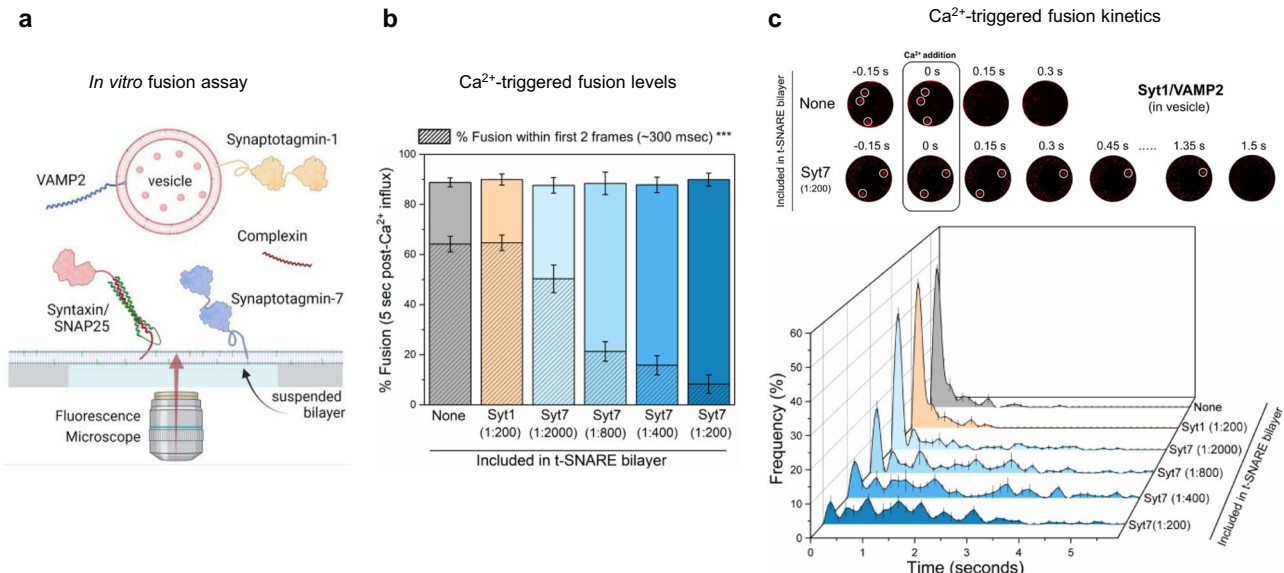

**Fig. 1 | Impact of Syt7 on $Ca^{2+}$-evoked fusion of Syt1/VAMP2 vesicles. a** In a typical in vitro fusion experiment, vesicles containing VAMP2 (~70 copies) and Syt1 (~20 copies) were added to a suspended bilayer membrane (formed on a silicon substrate with 5 μm holes) reconstituted with Syntaxin/SNAP25 (1:400 protein-to-lipid ratio) ±Syt7 in the presence of Complexin (2 μM) in solution. The fate of each vesicle before and after the addition of 100 μM $Ca^{2+}$ was monitored by a confocal microscope using a fluorescent (ATTO647N) marker included in the vesicle. **b** Syt7 (included in the t-SNARE containing bilayer) had no impact on the fusion competence of docked Syt1/VAMP2 vesicles, with ~85% fusing within 5 s after the arrival of 100 μM $Ca^{2+}$ signal at or near the docked vesicles. The hatched bar represents the percent fusion occurring with 2 frames (~300 ms) following $Ca^{2+}$ arrival. **c** Syt7 altered the $Ca^{2+}$-triggered fusion kinetics of docked Syt1/VAMP2 vesicles. Top, Representative time-lapse image of $Ca^{2+}$-evoked fusion of docked vesicles shows that without Syt7 the vesicles fuse rapidly and synchronously following $Ca^{2+}$

addition. The inclusion of Syt7 (1:200 protein-to-lipid ratio) introduces variable delays in $Ca^{2+}$-evoked fusion kinetics. Individual vesicles (white circles) docked within 5 μm suspended bilayer are shown. Bottom, quantitative analysis of $Ca^{2+}$-evoked fusion of Syt1/VAMP2 vesicles introduces a concentration-dependent delay in the $Ca^{2+}$-evoked fusion kinetics, resulting in a significant reduction in the proportion of vesicles fusing within the first 2 frames (~300 ms) following $Ca^{2+}$ arrival at time t = 0 s. Data (mean ± standard deviation) are from 5 independent experiments ($N = 5$) for each condition (~40–50 vesicles per experiment). One-way ANOVA revealed statistically significant difference (***$p < 0.001$) in $Ca^{2+}$-coupled fusion occurring within ~300 ms (hatched bar) between groups. The data from ANOVA and Tukey's HSD post-hoc comparing specific groups is shown in Supplementary Table 1. The source data is provided as a 'Source Data' file. Figure 1a created in BioRender. Krishnakumar, S. (2023) BioRender.com/f92r389.

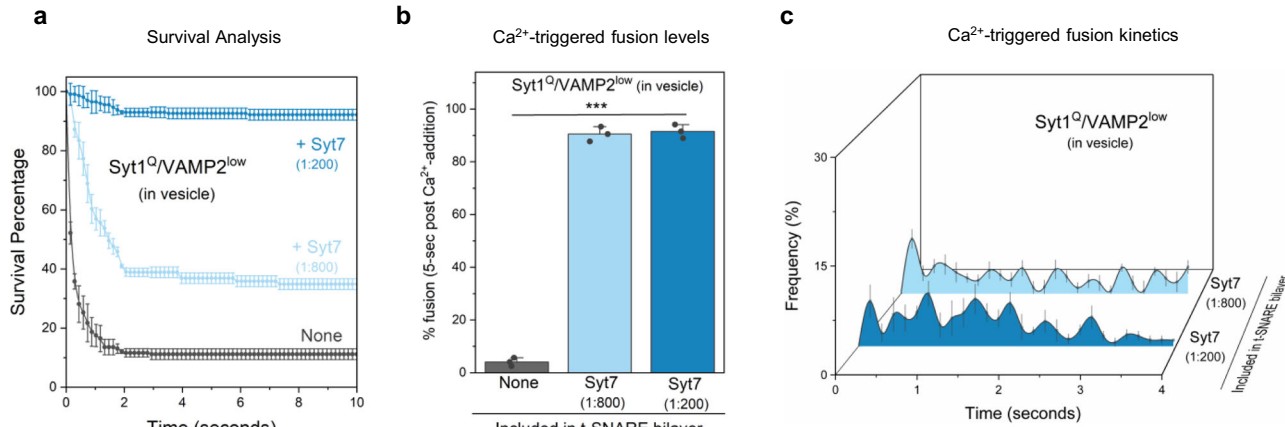

**Fig. 2 | Contribution of Syt7 to the establishment of the fusion clamp.** The involvement of Syt7 in the fusion clamp was evaluated using vesicles containing low-copy VAMP2 (~15 copies) and a non-clamping Syt1 mutant, Syt1$^Q$ (carrying R281A,E295A,Y338W,R398A,R399A mutations that disrupt the Syt1-SNARE primary interface) in the absence of CPX. **a** The time between docking and spontaneous fusion was measured for each docked vesicle and the 'docking-to-fusion' latency time was cumulatively expressed as the survival percentage. This 'survival analysis' provided the measure of the strength of the fusion clamp. In the absence of Syt7 (gray), the majority of the docked VAMP2$^{low}$/Syt1$^Q$ vesicles proceed to fuse spontaneously with a half-time of ~1 s. The inclusion of Syt7 in the bilayer resulted in stably docked vesicles in an immobile state, with clamping efficiency correlating with the amount of Syt7 included. Approximately 40% of vesicles were clamped under low Syt7 concentration (1:800, light blue) and this increased to ~90% under high Syt7 concentration (1:200, dark blue). **b, c** Syt7 clamped VAMP2$^{low}$/Syt1$^Q$ vesicles remained fusion competent and could be triggered to fuse by the addition of Ca$^{2+}$ (100 μM) and the observed fusion was desynchronized to the Ca$^{2+}$ signal. In the absence of Syt7, a very small percent of the docked VAMP2$^{low}$/Syt1$^Q$ vesicles underwent fusion which precluded meaningful kinetic analysis. Data (mean ± standard deviation) are from 3 independent experiments ($N$ = 3) for each condition (~40–50 vesicles per experiment). One-way ANOVA revealed statistically significant difference (***$p$ < 0.001) in % Ca$^{2+}$-evoked fusion of docked vesicles in the presence of Syt7 as compared to the condition without Syt7 in the bilayer. The data from ANOVA and Tukey's HSD post-hoc comparing specific groups are shown in Supplementary Table 2. The source data is provided as a 'Source Data' file.

1:800 and 1:200 Syt7$^{WT}$ respectively) (Fig. 3b). Notably, the observed fusion was desynchronized from the Ca$^{2+}$ signal (Fig. 3b). Taken together, these data suggest that Syt1 and Syt7 act on the same vesicles, likely targeting the same SNARE complexes, and their cooperative action in regulating Ca$^{2+}$-evoked fusion stems from a competitive binding of Syt1 and Syt7 to the same SNARE complex. Furthermore, these results unequivocally demonstrate that Syt1 acts as a 'fast' Ca$^{2+}$-sensor to trigger rapid Ca$^{2+}$-evoked vesicle fusion, whereas Syt7 functions as a 'slow' Ca$^{2+}$-sensor that mediates release over longer time intervals.

Subsequently, we employed a quantitative pull-down assay to directly test the competitive interaction between Syt1 and Syt7 with the same SNARE complex. While the binding of Syt1 to Syntaxin/SNAP25 is well-documented[34–36], the Syt7-SNARE interaction remains poorly understood. Hence, we initially conducted a pull-down experiment using Syt7 immobilized on agarose beads as 'bait' and pre-formed CPX-SNARE complex at varying concentrations as the 'prey'. Western-blot analysis confirmed direct molecular interaction between the CPX-SNARE complex and Syt7, revealing a saturable dose-response curve with an estimated apparent affinity (K$_d$) ~20 μM (Supplementary Fig. 4). We then examined the binding of 30 μM CPX-SNARE complex to Syt7-coated beads in the presence of varying concentrations (ranging from 1 μM to 50 μM) of Syt1. The inclusion of Syt1 disrupted the Syt7-SNARE interaction, resulting in near complete abrogation of binding at Syt1 concentrations ≥ 30 μM (Fig. 3c). This analysis directly demonstrates the competitive nature of the binding between Syt1 and Syt7 to the SNARE complex. In summary, our data argue that the kinetics of Ca$^{2+}$-triggered fusion are governed by the number of Syt1 or Syt7 associated SNAREpins, which is in turn determined by the relative abundance of these two proteins.

**Differential clamp removal rates of Syt1 and Syt7 shape Ca$^{2+}$ triggered fusion kinetics**
How do Syt1 and Syt7 shape the kinetics of vesicular fusion? It has been proposed that the cooperative action of Syt1 and Syt7 in regulating vesicular release can be explained by a 'release of inhibition' model[10–12].

According to this model, Syt1 and Syt7 along with CPX bind to SNAREpins at docked SVs and clamp vesicular fusion at rest. Ca$^{2+}$ activation of Syt1 and Syt7 leads to the release of the fusion clamp. Thereby, the rate of the Ca$^{2+}$-triggered removal of the fusion clamp determines the overall efficacy and kinetics of SV fusion (Fig. 4a). The model further posits that the differential Ca$^{2+}$/membrane binding properties of Syt1 and Syt7, along with the relative numbers of Syt1 or Syt7 bound SNAREs on a given vesicle, fine-tune the release properties in response to Ca$^{2+}$ signals. Indeed, our experimental data with Ca$^{2+}$-insensitive Syt1$^{DA}$ and Syt7$^{DA}$ mutants support the 'release of inhibition' model, as both mutants blocked the Ca$^{2+}$-triggered fusion of docked vesicles, consistent with the model predictions (Fig. 3).

To investigate whether the differences in Ca$^{2+}$/membrane binding properties of Syt1 and Syt7 could explain our current results, we adapted the previously developed computational framework[12]. This modeling framework enables us to simulate SV fusion in response to specific Ca$^{2+}$ signals for different Synaptotagmin fusion clamp architectures. Drawing on structural studies, within the default model, we assumed that each vesicle contains six SNARE complexes[37] and each SNARE complex can bind two Syt1 molecules[34]. Additionally, we postulated that Syt7 might compete with Syt1 for one of these binding sites. Consequently, we considered two limiting cases for the fusion clamp's architecture: either Syt1/Syt1 or Syt1/Syt7 (Fig. 4a). These scenarios correspond to experimental conditions without Syt7 or with a saturating level of Syt7 in the lipid bilayer respectively. As in our previous work[12], we assumed that Ca$^{2+}$ binding and membrane loop insertion of the C2B domain of Syt1 or C2A domain of Syt7 leads to the instantaneous removal of the fusion clamp (Fig. 4b, Scheme 1). The release of the clamp enables the full zippering of freed SNAREs, and each SNARE complex independently contributes towards lowering the fusion barrier, thereby catalyzing SV fusion.

As a model input, we incorporated experimentally estimated changes in [Ca$^{2+}$] at the lipid bilayer, corresponding to a ramped increase of [Ca$^{2+}$] from 0 to 100 μM (Supplementary Fig. 3). In the absence of Syt7 (Syt1/Syt1 clamp), the standard model closely reproduced the kinetics of vesicular fusion in response to the addition of

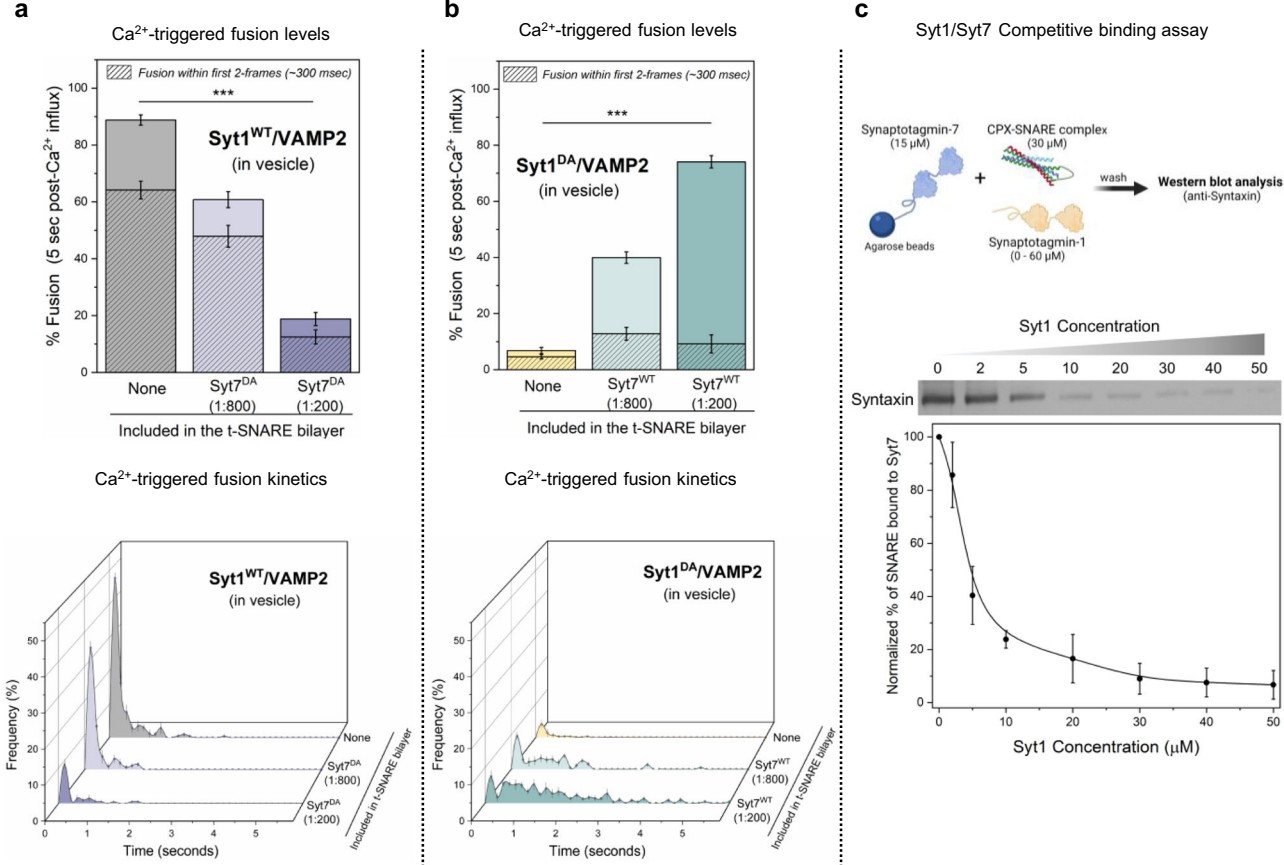

**Fig. 3 | Syt1 and Syt7 synergistically regulate Ca²⁺-evoked fusion via competitive binding to the same SNARE complex. a** The inclusion of the Ca²⁺ binding deficient Syt7 mutant (Syt7$^{DA}$) in the bilayer inhibited Ca²⁺ (100 μM) evoked fusion of Syt1$^{WT}$/VAMP2 vesicles in a dose-dependent manner, without altering the overall fusion kinetics. **b** Syt7$^{WT}$ from the bilayer rescued the Ca²⁺-evoked fusion of Syt1$^{DA}$/VAMP2 vesicles but the fusion events were desynchronized to the Ca²⁺ signal. Complexin (2 μM) in solution was included in all experiments. **c** Quantitative pull-down and Western-blot analysis with Syt7 as 'bait' and CPX-SNARE complex as 'prey' demonstrate that Syt1 disrupts Syt7-SNARE interaction in a concentration-

dependent manner. Data (mean ± standard deviation) are from 5 independent experiments ($N = 5$) for each condition (~40–50 vesicles per experiment) in (**a**) and (**b**) and from 4 independent experiments ($N = 4$) in (**c**). One-way ANOVA revealed statistically significant difference in % Ca²⁺-evoked fusion of docked vesicles in the presence of Syt7$^{DA}$ (***$p < 0.001$) or Syt7$^{WT}$ (***$p < 0.001$) as compared to condition without Syt7 in the bilayer. The data from ANOVA and Tukey's HSD post-hoc comparing specific groups are shown in Supplementary Tables 3 and 4 respectively. The source data is provided as a 'Source Data' file. Figure 3c (top) created in BioRender. Krishnakumar, S. (2023) BioRender.com/p26b995.

Ca²⁺ (Fig. 4c). Indeed, the time course of Syt1-mediated vesicular fusion closely follows the kinetics of the [Ca²⁺] signal (Fig. 4c). This suggests that Ca²⁺ diffusion is likely the rate-limiting step governing vesicular fusion kinetics under our experimental conditions. However, this model failed to replicate the slower vesicular fusion kinetics observed when Syt7 was included (Syt1/Syt7 clamp).

Under our experimental conditions, Syt1 and Syt7 are predicted to exhibit comparable Ca²⁺-activation patterns, with Syt7 being slightly more sensitive than Syt1 (Supplementary Fig. 5). This suggests that the Ca²⁺-triggered membrane insertion of Syt7 is not the rate-limiting step in the removal of the Syt7 fusion clamp. Consequently, we adapted the model to include a delay between the Ca²⁺-triggered membrane insertion of the Syt7 C2A domain and the removal of the fusion clamp (Fig. 4b, Scheme 2). This modification allowed us to reconcile the model with the experimental data under Syt1/Syt7 clamp conditions (Fig. 4c).

Given the ongoing debate surrounding the exact number of SNARE complexes on an RRP vesicle[37,38], as well as the architecture of the Synaptotagmin fusion clamp[30,34,35], we explored alternative fusion clamp configurations. Specifically, we varied the number of SNAREpins in the vesicles from six to twelve and examined scenarios where a single Synaptotagmin molecule - either Syt1 or Syt7 - could bind to and clamp an individual SNAREpin (Supplementary Fig. 6). As expected,

increasing the number of SNAREpins, or using a single clamp instead of a dual clamp accelerated fusion rates at low [Ca²⁺]. However, at saturating [Ca²⁺], the models' outputs closely aligned with those of the default dual-clamp model with six SNAREpins (Supplementary Fig. 6). Taken together, our data argue that the mechanisms of clamp removal are distinct for Syt1 and Syt7 and that the differing rates of clamp removal – rapid for Syt1 and slower for Syt7 – are key factors determining the Ca²⁺ triggered vesicular fusion kinetics.

## Syt7 enhances Ca²⁺ synchronized fusion under elevated [Ca²⁺]$_{basal}$ conditions

In addition to modulating fusion kinetics, Syt7 has also been implicated in the facilitation of synchronous neurotransmitter release during neuronal activity[17,18]. This short-term plasticity of vesicular release has been linked to the accumulation of [Ca²⁺]$_{residual}$ in low micromolar range within the presynaptic terminal due to sustained neuronal activity[1,15,16]. Hence, we investigated the effect of elevated basal [Ca²⁺]$_{basal}$ on Ca²⁺-triggered release properties of Syt1/VAMP2 vesicles in the absence and presence of Syt7.

The inclusion of 0.5 μM [Ca²⁺]$_{basal}$ during the vesicle docking phase had little to no effect on vesicle docking or clamping (i.e., spontaneous fusion) of vesicles across all conditions tested (Supplementary Fig. 7). It also did not affect vesicle fusion triggered by 100 μM

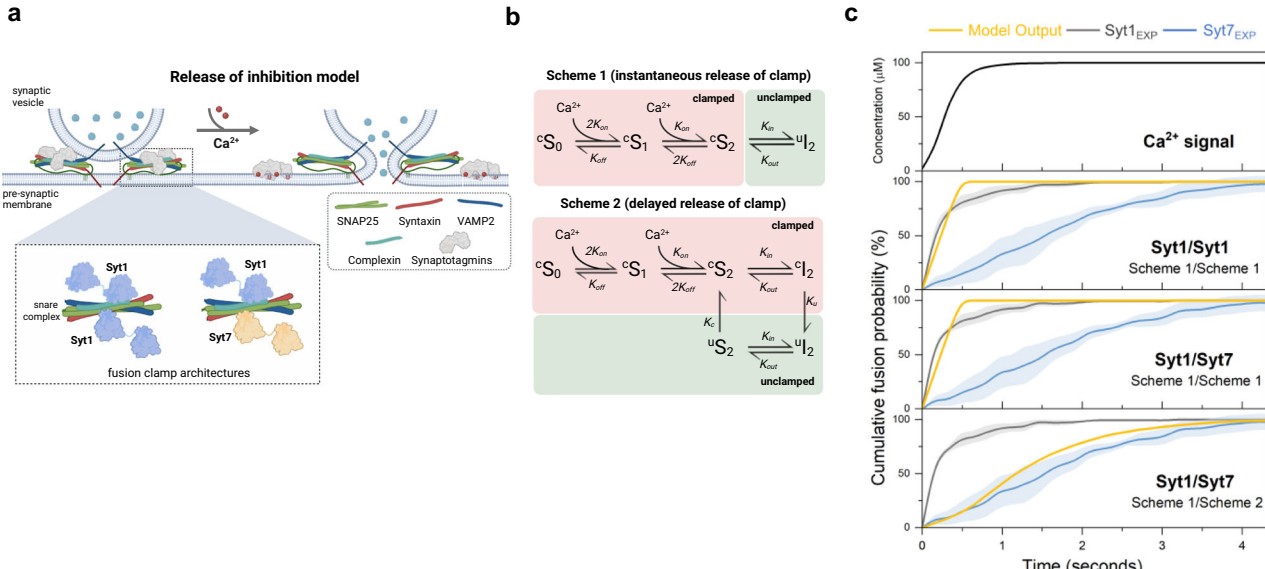

**Fig. 4 | Computational model of synergistic activation of vesicular fusion by Syt1 and Syt7. a** Schematic illustration of the release of inhibition model. At rest, fusion of vesicles is inhibited ('clamped') by binding of Syt1 and Syt7 along with CPX to partially assembled SNAREpins. Upon $Ca^{2+}$ binding, the C2 domains of Syt1 and Syt7 insert into the membrane, leading to the removal of the fusion clamp. This allows the complete zippering of the SNARE complexes, resulting in vesicular fusion. Inset shows that two clamp architectures are considered in the default model: dual Syt1/Syt1 or dual Syt1/Syt7 clamp (see Supplementary Fig. 6 for additional clamp architectures tested). **b** Kinetic reaction schemes describing $Ca^{2+}$-triggered release of the fusion clamp. Each modeled C2 domain sequentially binds two $Ca^{2+}$ ions which triggers the insertion of its aliphatic loop into the membrane. Scheme 1 assumes that membrane insertion results in the instantaneous removal of the Synaptotagmin fusion clamp, while Scheme 2 assumes a delay between membrane insertion and the removal of the clamp. $S_0$, $S_1$, $S_2$ refer to 0, 1 or 2 $Ca^{2+}$ bound

state of the C2 domains, while $I_2$ refers to membrane inserted state of the $Ca^{2+}$-bound C2 domain. The prefixes $^c$ and $^u$ refer to the 'clamped' or 'unclamped' state respectively. **c** The time course of vesicular fusion (Model Output) simulated in response to the experimentally constrained $Ca^{2+}$ signal (Supplementary Fig. 3) for models with different clamp architecture and kinetics of clamp reversal. Experimental data (mean ± standard deviation from Fig. 1c) for the $Ca^{2+}$-triggered fusion of Syt1 containing vesicles in the absence ($Syt1_{EXP}$) or the presence of saturating levels of Syt7 ($Syt7_{EXP}$) are plotted for comparison. The model suggests that experimentally observed fusion kinetics can be explained by the mechanism with differential rates of fusion clamp removal for Syt1 (instantaneous) and Syt7 (delayed). For each modeled condition a minimum of 1000 stochastic simulations were performed to calculate the average response. The source data is provided as a 'Source Data' file. Figure 4a created in BioRender. Krishnakumar, S. (2023) BioRender.com/x49d271.

[Ca²⁺] under control (no Syt7 in the bilayer) conditions (Fig. 5a). However, when Syt7 was included in the bilayer (at 1:200 protein-to-lipid ratio), it enhanced the fast component of $Ca^{2+}$-evoked release (within the first 300 milliseconds after the initial arrival of $Ca^{2+}$ signal), increasing it from ~8% to ~35%, without changing the overall level of fusion over the 5-second interval (Fig. 5a). Likewise, the computational model incorporating a delay in the removal of the Syt7 fusion clamp (Scheme 2) also replicated the synchronization of vesicular fusion when Syt7 was pre-activated with low micromolar [Ca²⁺] (Fig. 5b). We only tested 0.5 μM [Ca²⁺]$_{basal}$ in our in vitro assay because higher [Ca²⁺]$_{basal}$ significantly increased spontaneous fusion of docked vesicles, preventing meaningful analysis. However, the degree of synchronization predicted by the computational model correlated with the concentration of $Ca^{2+}$ utilized for pre-activation (Fig. 5b). Together, these data show that when pre-activated by low micromolar [Ca²⁺]$_{basal}$, Syt7 enhances $Ca^{2+}$-synchronized vesicle fusion.

Interestingly, disrupting $Ca^{2+}$-binding to Syt1 (Syt1$^{DA}$) did not abolish the Syt7-dependent synchronization of vesicular release with elevated [Ca²⁺]$_{basal}$ (Fig. 5a). Indeed, we observed a similar proportion of $Ca^{2+}$-synchronized release between Syt1$^{WT}$/Syt7 and Syt1$^{DA}$/Syt7 conditions (Fig. 5a). This indicates that when primed by elevated [Ca²⁺]$_{basal}$, Syt7 is capable of independently mediating fast $Ca^{2+}$-synchronized release.

## Discussion

Our study presents one of the first in vitro reconstitution of different modes of $Ca^{2+}$-triggered SV fusion with minimal protein components. We demonstrate that Syt1 and Syt7, along with SNAREs and CPX, are sufficient to recapitulate fast and delayed $Ca^{2+}$-evoked vesicular fusion

as well as $Ca^{2+}$-dependent facilitation of vesicular release. Our data show that under resting conditions, Syt1, Syt7, CPX work together to arrest SNARE assembly and produce stably, docked RRP-like vesicles. When activated by the $Ca^{2+}$ signal, the Syt1 triggers the fast component, whilst Syt7 drives the slow component of the resultant vesicular fusion. Mutational analysis with $Ca^{2+}$-insensitive Syt1 and Syt7 mutants indicates that the synergistic action of Syt1 and Syt7 in the regulation of $Ca^{2+}$-evoked vesicular fusion can be described by the 'release of inhibition' model, which posits that the rate of vesicular fusion is governed by the release of the Syt1 and/or Syt7 fusion clamp upon their $Ca^{2+}$ activated membrane insertion[10–12]. Furthermore, we observe that the kinetics of Syt7-mediated vesicular fusion are significantly influenced by [Ca²⁺]$_{basal}$. We demonstrate that Syt7 is capable of mediating fast, $Ca^{2+}$-synchronized release, independent of Syt1, when [Ca²⁺]$_{basal}$ is elevated into low micromolar range. We posit that this phenomenon underlies the critical role of Syt7 in short-term facilitation of fast synchronous release component during sustained neuronal activity[1,4].

The temporal resolution of our fusion assay is limited by the shape of the $Ca^{2+}$ signal, which ramps up to 100 μM within approximately 750 milliseconds. Indeed, we find that the rate of fast Syt1-mediated fusion closely mirrors the rate of [Ca²⁺] increase at the lipid bilayer. Due to technical limitations, our in vitro assay cannot replicate the rapid, millisecond-scale $Ca^{2+}$ transients evoked by action potentials at the presynaptic active zone. Hence, we utilized a computational modeling framework, capable of simulating vesicular fusion kinetics in response to specific [Ca²⁺] transients, to relate our findings to neurotransmitter release kinetics in neuronal synapses. The computational implementation of the 'release of inhibition' model with the experimental

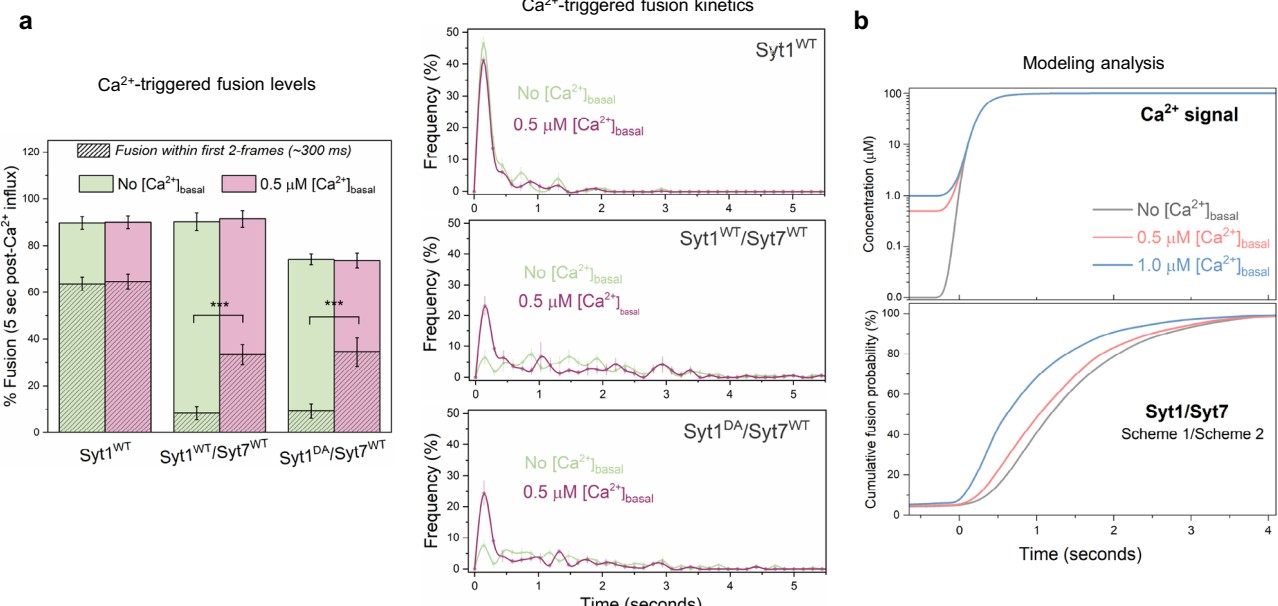

**Fig. 5 | Syt7 enhances Ca²⁺-synchronized fusion under elevated [Ca²⁺]basal conditions. a** Comparison of the Ca²⁺ (100 μM) evoked fusion characteristics without (green) or with (pink) 0.5 μM [Ca²⁺]basal included during vesicle docking reveals that when pre-activated, Syt7 (included in the bilayer at 1:200 protein-to-lipid ratio) increases the proportion of Ca²⁺-coupled release of Syt1-containing vesicles (Syt1ᵂᵀ/Syt7ᵂᵀ) without changing the overall fusion levels. This enhancement was not observed with Syt1ᵂᵀ alone. Notably, a similar degree of enhancement of Ca²⁺-synchronized release was observed with vesicles containing Ca²⁺-binding deficient Syt1ᴰᴬ (Syt1ᴰᴬ/Syt7ᵂᵀ). Data (mean ± standard deviation) are from 5 independent experiments ($N = 5$) for each condition (~40–50 vesicles per experiment). Complexin (2 μM) in solution was included in all experiments. **b** The dual Syt1/Syt7 clamp model, incorporating the delayed release of the clamp for Syt7, reproduces the experimentally observed enhancement of synchronous release following pre-activation with low micromolar [Ca²⁺]. The extent of facilitation correlated with the level of pre-activating [Ca²⁺] used. For each modeled condition a minimum of 1000 stochastic simulations were performed to compute the average response. ***$p < 0.001$ using the one-sided students' t-test comparison to the control condition with no [Ca²⁺]basal. The source data is provided as a 'Source Data' file.

ramped [Ca²⁺] signal as an input, closely reproduced the kinetics of Syt1-mediated vesicular fusion observed in our reconstituted fusion assay. Notably, the same model implementation reproduced the millisecond kinetics of vesicular fusion in response to fast [Ca²⁺] transients observed in live synapses[12]. This indicates that the reconstituted fusion assays replicate the functionality of the component proteins in living synapses, with comparable operational efficacy. Thus, our data strongly suggest that Syt1 and Syt7 are likely sufficient to describe the synchronous and asynchronous components of AP-evoked neurotransmitter release in neuronal synapses.

The computational modeling further suggests that the differential effects of Syt1 and Syt7 on vesicular release kinetics can be attributed to the differential strength and kinetics of Ca²⁺-triggered reversal of their respective fusion clamps. Specifically, our data indicate that activation leads to almost instantaneous removal of the Syt1 fusion clamp but delayed release of the Syt7 clamp. Both Syt1 and Syt7 are expected to bind and clamp the SNARE complexes via their C2B domains[34,36]. However, the critical distinction in their roles in the regulation of SV fusion arises from the Ca²⁺ activation of the Syt1 C2B domain compared to the Syt7 C2A domain[19]. The fast removal of the Syt1 clamp may be attributed to the rapid dissociation of Syt1 from the SNARE complex at the primary interface upon Ca²⁺-triggered membrane insertion of its C2B domain, as demonstrated in biochemical and structural studies[35,36]. In contrast, for Syt7 the decoupling of SNARE binding (C2B domain) and Ca²⁺ activation (C2A domain) may contribute to the slower disassembly of the Syt7 fusion clamp.

While our data indicates that Syt1, Syt7, and CPX all are involved in establishing the fusion clamp, the precise molecular composition of the fusion clamp on a RRP vesicle remains unknown. In particular, the clamping function of CPX and Syt7 remains a topic of debate. For example, genetic removal of CPX potentiates spontaneous events in invertebrate model systems[39,40], but acute removal of CPX in cultured

mouse neurons abates both spontaneous and evoked neurotransmitter release[41] suggesting that CPX is principally a positive regulator of fusion in mammalian synapses. However, a recent study showed that CPX mutants that disrupt the clamping function under in vitro conditions[26,42] selectively potentiate spontaneous neurotransmitter release, while leaving the evoked release largely untouched[42]. This suggests that the inhibitory function of CPX might be normally masked by a more pronounced positive function in mammalian synapses. Similarly, in central synapses the genetic deletion of Syt7 does not alter frequency of spontaneous release[19]. However, the overexpression of Syt7 reverses the increased mini frequency observed in Syt1 knockout neurons[19]. This suggests that Syt7 can substitute for Syt1 in clamping mini release, but its clamping role may not be apparent under normal physiological conditions due to low expression levels of Syt7 in presynaptic terminals. Additional research is needed to delineate the precise molecular organization of the prefusion 'clamped' state and the mechanisms of Ca²⁺-triggered reversal of the fusion clamp.

Additionally, we find that Syt1 and Syt7 compete to bind the same SNARE complexes. Consequently, the relative abundance of these Ca²⁺ sensors shape the kinetics and plasticity of Ca²⁺-evoked vesicular fusion in our assay. This observation offers a mechanistic explanation for physiological data demonstrating that the relative expression levels of Syt1 and Syt7 regulate both the kinetics and plasticity of neurotransmitter release in neuronal synapses[43,44]. The number of Syt1 per vesicle is tightly controlled (~15–20 copies per SV) and local Syt1 concentration under a docked vesicle is estimated to be in the range of 5–10 mM[45,46]. Syt1 interacts with SNARE complex at the primary interface with an affinity of ~10 μM, suggesting a near-complete saturation of Syt1-SNARE binding under physiological conditions. However, the precise concentration of Syt7 is unknown and varies across synapses. Consequently, the number of Syt1- or Syt7-bound SNAREpins under a

given RRP vesicle is likely determined by local abundance of Syt7. This implies that controlling the local abundance of Syt7 molecules could be a straightforward mechanism by which synapses can dynamically adjust the strength and efficacy of synaptic transmission during sustained activity.

While the precise mechanism of Syt7-mediated facilitation is not fully understood, it is hypothesized that activation of Syt7 could enhance the release by two different mechanisms: (i) by increasing the probability of RRP vesicles and/or (ii) by enhancing the activity-dependent docking of SVs[4,47]. Our in vitro reconstitution experiments and modeling demonstrate that, when pre-activated, Syt7 enhances the synchronous release of pre-docked vesicles. Our previous computational modeling study provided insights into the possible molecular mechanisms underlying Syt7-mediated facilitation[12]. It revealed that the rate of vesicle fusion is dictated by the time taken for three SNAREpins to be released from the fusion clamp on a specific vesicle. Due to its high $Ca^{2+}$/membrane affinity, Syt7 is partially activated at low micromolar $[Ca^{2+}]$, weakening the fusion clamp. This in turn, accelerates the liberation of three SNAREpins upon the arrival of $[Ca^{2+}]$ signal, thereby enhancing the fast release component. Consistent with this hypothesis, we observed a comparable level of facilitation when $[Ca^{2+}]_{basal}$ levels were elevated either during or after the vesicles reached the immobile clamped state (Supplementary Fig. 7).

We note that, apart from the distinct $Ca^{2+}$-release sensors, other proteins and mechanisms also play a role in regulating the timing and plasticity of neurotransmitter release. For example, the genetic deletion of Syt7 does not fully eliminate asynchronous release or short-term facilitation in some synapses[43,48]. Consequently, it is suggested that other $Ca^{2+}$-release sensors (e.g., Syt1, Syt3 and Doc2A) may also contribute to the asynchronous release component[21–23,49]. Additionally, short-term facilitation of synchronous neurotransmitter release may result from enhanced $Ca^{2+}$ transients at release sites, driven by the saturation of $Ca^{2+}$ buffers during repetitive activity[25,50]. Furthermore, the strength and efficacy of neurotransmitter release is also regulated by $Ca^{2+}$-dependent SV docking, priming, and recycling. As such, SV priming factors (e.g. RIM, Munc13) could also influence neuro-transmitter release dynamics[9,51]. Nonetheless, our results highlight the central role of Syt1 and Syt7 in decoding presynaptic $Ca^{2+}$ dynamics and translating this into complex patterns of vesicular release.

## Methods

### Proteins & materials
In this study, we used the following clones that have been described previously[26,30] including full-length VAMP2 (human VAMP2-His[6], residues 1–116); full-length t-SNARE complex (mouse His[6]-SNAP25B, residues 1–206 and rat Syntaxin1A, residues 1–288); CPX (human His[6]-Complexin 1, residues 1–134); Syt1 wild-type (rat Synaptotagmin1-His[6], residues 57–421) and mutants (D309A, D 363A, D365A; Syt1[DA]) and (R281A, E295A,Y338W,R398A,R399A, Syt1[Q]) in the same background.

Our initial experiments were conducted with the full-length Syt7 wild-type protein (His[6]-SUMO-rat Synaptotagmin-7, residues 17–403). However, this construct posed technical challenges due to its low and highly variable membrane reconstitution efficiency. Hence, we modified the construct by adding a second transmembrane domain (TMD) from Syt1 with a flexible 16 residue GSGS linker, resulting in His[6]-SUMO-Syt1[TMD]-Syt7 construct (referred to as Syt7[WT] in this manuscript). The inclusion of Syt1[TMD] (in addition to the Syt7[TMD]) improved the reconstitution efficiency of the Syt7[WT] protein into the membrane, while the flexible GSGS linker ensured the proper orientation of the two TMDs and Syt7 C2AB domains (Supplementary Fig. 1). Control experiments showed that effect of Syt7, whether containing one or two TMDs, on $Ca^{2+}$-evoked fusion of Syt1/VAMP2 vesicles were indistinguishable (Supplementary Fig. 1).

We also generated Syt7 mutant (D225A, D227A, D233A, D357A, D359A; Syt7[DA]) in the same background. We purchased the cDNA to produce the SUMO nanobody (nanoCLAMP SMT3-A1) from Nectagen (Lawrence, KS). The lipids used in the study, including 1,2-dioleoyl-snglycero-3-phosphocholine (DOPC), 1,2-dioleoyl-sn-glycero-3-(phospho-L-serine) (DOPS), and phosphatidylinositol 4, 5-bisphosphate (PIP2) were purchased from Avanti Polar Lipids (Alabaster, AL). ATTO647N-DOPE and ATTO465-DOPE were purchased from ATTO-TEC, GmbH (Siegen, Germany) and Calcium Green conjugated to a lipophilic 24-carbon alkyl chain (Calcium Green C24) was purchased from Abcam (Cambridge, UK). All other research materials and consumables, unless specified, were purchased from Sigma-Aldrich (St Louis, MO) and Thermo Fisher Scientific (Waltham, MA)

### Protein expression and purification
All proteins were expressed and purified in a bacterial expression system as described previously[26,30] (Supplementary Fig. 1). In summary, proteins were expressed in *E. coli* BL21(DE3) cells (Novagen, Madison, WI) under 0.5 mM IPTG induction for 4 h. Bacterial cells were pelleted and then lysed using a cell disruptor (Avestin, Ottawa, Canada) in lysis buffer containing 25 mM HEPES, 400 mM KCl, 4% Triton X-100, 10% glycerol, pH 7.4 with 0.2 mM Tris[2-carboxyethyl] phosphinehydrochloride (TCEP), and EDTA-free Complete protease inhibitor cocktail (Merck, Rahway, NJ). The resulting lysate was clarified using a 45Ti rotor (Beckman Coulter, Atlanta, GA) at 40,000 RPM for 30 min and subsequently incubated with pre-equilibrated Ni-NTA resin overnight at 4 °C. The resin was washed with wash buffer containing 25 mM HEPES pH 7.4, 400 mM KCl, 0.2 mM TCEP. The wash buffer was supplemented with 1% octylglucoside (OG) for Syt1 and SNARE, and with 0.2% Triton-X-100 for Syt7. Proteins were eluted from beads using 400 mM Imidazole and their concentrations were determined using a Bradford Assay (BioRad, Hercules, CA) with BSA standard. The Syt1 and Syt7 proteins were further treated with Benzonase (Millipore Sigma, Burlington, MA) at room temperature for 1 h with Syt1 additionally being run through ion exchange (Mono S) to remove DNA/RNA contamination. SDS-PAGE analysis was done to check the purity of the proteins, and all proteins were flash-frozen in small aliquots and stored at −80 °C with 10% glycerol.

### Vesicle preparation
Small unilamellar vesicles containing VAMP2 and Syt1 were prepared using rapid detergent dilution and dialysis method, followed by additional purification on discontinuous Optiprep gradient by ultracentrifugation[26,30]. To mimic synaptic vesicle lipid composition, we used 88% DOPC, 10% DOPS, and 2% ATTO647N-PE, with the protein-to-lipid input ratio of 1:100 for VAMP2 for physiological density, 1:500 for VAMP2 at low copy number, and 1:250 for Syt1. Informed by previous work[26,30] that characterized the reconstitution efficiency and inside/outside ratio of these proteins, we estimate the vesicle contains ~70 copies of outside facing VAMP2 and ~20 copies of outside facing Syt1 (at physiological conditions) and ~15 copies of VAMP2 and ~20 copies of Syt1 (under low VAMP2 conditions).

### Suspended lipid bilayer formation
To form the suspended lipid bilayer, we first prepared giant unilamellar vesicles (GUVs) containing t-SNARE ± Syt7 were prepared using the osmotic shock protocol as described previously[52]. To mimic the presynaptic plasma membrane, the lipid composition of the GUVs was 80% DOPC, 15% DOPS, 3% PIP2%, and 2% ATTO465-PE. The t-SNARE complex (1:1 Syntaxin/SNAP25) was included at the protein-to-lipid input ratio of 1:200 to yield a final concentration of 1:400. Incorporating the t-SNARE complex enabled us to circumvent the necessity for the SNARE-assembling chaperones Munc18 and Munc13[53]. When warranted, Syt7 was added at a protein-to-lipid input ratio of 1:50, 1:100, 1:200, and 1:1000 to yield the defined concentrations of Syt7 tested, based on the reconstitution efficiency (~50%) and 50-50 inside/outside

ratio determined by protease (Chymotrypsin) accessibility assay (Supplementary Fig. 1).

Subsequently, t-SNARE (±Syt7) containing GUVs were burst on freshly plasma-cleaned Si/SiO2 chips decorated with a regular array of 5 μm diameter holes in HEPES buffer (25 mM HEPES, 125 mM KCl, 0.2 mM TCEP, 5 mM MgCl2 pH 7.4). The bilayers were then extensively washed with the same HEPES buffer containing 1 mM MgCl2. For each experiment, the fluidity of the bilayers was verified using FRAP of the Atto-465 fluorescence (Supplementary Fig. 8). As a control, we tested and confirmed that the mobility of Alexa488 labeled t-SNAREs is not affected by the inclusion of Syt7 (Supplementary Fig. 8).

### Single vesicle fusion assays

The vesicle docking and fusion experiments were carried out as described previously[26,28,30]. Typically, in each experiment, approximately 100 nM lipids worth of vesicles, along with CPX (2 μM final concentration) were added using a pipette and then allowed to interact with the suspended bilayer for 5 mins. ATTO647N-PE fluorescence was used to track the fate of individual vesicles, i.e. vesicle docking, post-docking diffusion, docking-to-fusion delays, and spontaneous fusion events. Docked immobile vesicles that remained un-fused during the initial 10 min observation period were defined as 'clamped'. Fusion was identified as a sharp, rapid decrease in fluorescence intensity, as the lipids from the vesicles diffused into the bilayer. After the initial 5-minute observation period, the excess vesicles in the chamber were removed by buffer exchange, and 100 μM CaCl2 was added to quantify the Ca²⁺-triggered fusion of the pre-docked vesicles. To cover large areas of the planar bilayer and simultaneously record lipid mixing in large ensembles of vesicles (~40–50 per experiment), the movies were acquired at a speed of 147 ms per frame.

Ca²⁺ typically reached the vicinity of vesicles docked on the bilayer approximately 1-2 frames post-addition[26,30] and this correlated with the minima of the transmittance signal (Supplementary Fig. 3). For select experiments, we also included Calcium Green C24 in the bilayer to directly quantify the arrival of Ca²⁺ at the bilayer and confirmed that it matched with the transmittance signal change (Supplementary Fig. 3). As Calcium-green is a high-affinity Ca²⁺ sensor (Kd of ~100 nM), its fluorescence signal is typically saturated within a single frame following the arrival of Ca²⁺ at the bilayer (Supplementary Fig. 3). Hence, we utilized a soluble Alexa647 dye (~25 nM) mixed with 100 μM CaCl2 to track the diffusion of Ca²⁺ into the chamber. Assuming similar diffusion of Ca²⁺ and the Alexa647 dye, the changes in Alexa647 fluorescence provided a reliable indicator for estimating alterations in the [Ca²⁺] signal at or near the vesicles docked on the bilayer (Supplementary Fig. 3).

All experiments were carried out at 37 °C using an inverted laser scanning confocal microscope (Leica-SP5) equipped with a multi-wavelength argon laser including 488 nm, diode lasers (532 nm and 641 nm), and a long-working distance 40X water immersion objective (NA 1.1). The emission light was spectrally separated and collected by photomultiplier tubes.

### Pull-down binding analysis

To investigate the binding of Syt7 to SNAREs and assess the competitive binding of Syt1/Syt7 to the same SNARE complex, we used a pull-down analysis coupled with western-blot analysis. Briefly, we purified a SUMO-nanobody and covalently attached it to a CNBR-activated Sepharose resin. The nanobody-Sepharose resin was incubated (4 hr at 4 °C) with SUMO-Syt7 protein and subjected to extensive wash with HEPES buffer (50 mM HEPES, 400 mM KCl, 0.2 mM TCEP, 0.2% Triton-X-100, pH 7.4) to form the 'bait'. The CPX-SNARE complex was assembled and purified on the Superdex-200 column as described previously[54,55] in the HEPES buffer and used as the 'prey'. For the binding experiment, ~15 μM of Syt7-resin was incubated with pre-formed CPX-SNARE complex at varying (1–50 μM)

concentrations overnight at 4 °C with minimum agitation. The resin was washed extensively (5X) with HEPES buffer, followed by a stringent wash with HEPES buffer containing 1 M KCl to eliminate unbound proteins. The resin samples were subjected to SDS-PAGE gel electrophoresis, followed by western blotting using a Syntaxin monoclonal antibody (Abcam, Cambridge, UK) to quantify the amount of SNARE bound to the Syt7-resin. We used the same protocol for the competition assay, with the following modification: 15 μM Syt7-resin was incubated with 30 μM of CPX-SNARE complex, along with 1–60 μM Syt1 included in the solution overnight at 4 °C with minimum agitation.

### Computational modeling

Vesicular fusion in response to experimentally estimated changes in [Ca²⁺] was simulated using the computational modeling framework established in our previous work[12]. [Ca²⁺] stimulation profile was approximated based on the diffusion kinetics of Alexa 647 (Supplementary Fig. 3). Each RRP vesicle was associated with either six or twelve partially assembled SNAREpins which were clamped in this state by either one (Syt1 or Syt7) or two Synaptotagmins (Syt1/Syt1 or Syt1/Syt7). The dynamics of each Synaptotagmin C2 domain were described by the Markov kinetic schemes shown in Fig. 4b using the parameters we previously constrained. We assumed that $k_{on}$ was limited by diffusion to $1\,\mu M^{-1}\,ms^{-1}$ and $k_{off} = 150\,ms^{-1}$ based on the intrinsic Ca²⁺ affinity $K_d = 150\,\mu M$, which is similar for both Syt1 and Syt7 C2 domains[20,56,57]. $k_{in} = 100\,ms^{-1}$ based on the characteristic time for Synaptotagmin C2 domain rotation and membrane insertion[45]. $k_{out} = 0.67\,ms^{-1}$ for Syt1 and $k_{out} = 0.02\,ms^{-1}$ for Syt7 were determined from the apparent rates of C2 domain dissociation from lipid membranes ($k_{diss}$) measured in the presence of EGTA using stopped-flow experiments[57–59] as described in our previous work[12]. In reaction Scheme 1 (Fig. 4b), we considered that C2 domain membrane insertion leads to the instantaneous release of the fusion clamp. In reaction Scheme 2 (Fig. 4b), we assumed that a delay between the Ca²⁺-triggered membrane insertion of the Syt7 C2A domain and the removal of the fusion clamp is described by a first-order reaction with the rate $k_u = 0.0002\,ms^{-1}$ (this value was chosen as it best fits the experimental data within the tested $k_u$ range of $0.0001–0.01\,ms^{-1}$). We further assumed that the clamp could be restored following membrane dissociation at an identical rate $k_c = 0.0002\,ms^{-1}$. The rate of vesicular fusion was determined by assuming that the repulsive forces between a docked vesicle and the plasma membrane amount to an energy barrier $E_0 = 26\,k_BT$[60]. Once both of its Synaptotagmins are in an unclamped state a SNAREpin contributes $\Delta E = 4.5\,k_BT$ of work towards overcoming this energy barrier[61]. With $n$ uninhibited SNAREpins the fusion barrier is spontaneously overcome through thermal fluctuations at a rate given by the Arrhenius equation $R_{rate}(n) = A \cdot \exp(-\frac{E_0 - n\Delta E}{k_BT})$, where the pre-factor $A = 2.17 \times 10^9\,s^{-1}$ considering that a single SNARE complex can mediate fusion in vitro on a timescale of 1 sec[45,61]. Monte Carlo estimates of the cumulative probability of vesicle fusion in response to a Ca²⁺ activation signal were derived from at least 1000 stochastic simulations of individual vesicles in all scenarios. We estimate that this restricts prediction error due to stochastic variation to less than 1%. All simulations were carried out in MATLAB 2020b (The MathWorks Inc.) and Python 3.10. To accommodate the observed variability in the timing of Ca²⁺ signal arrival, which can span up to three frames under our experimental conditions (see Supplementary Fig. 3), we applied a temporal blurring of the model output, by smoothing the data across a time window of 0.45 s, equivalent to the duration of three imaging frames.

### Reporting summary

Further information on research design is available in the Nature Portfolio Reporting Summary linked to this article.

## Data availability

All relevant data that support the findings of this study has been included in the 'Source Data' file. Additional supporting information is available from the corresponding author upon request. Source data are provided with this paper.

## Code availability

The MATLAB and Python codes used in this study has been deposited in the GitHub repository: (https://github.com/ChrisAlexNorman/SytSim_Matlab); (https://github.com/ChrisAlexNorman/SytSim).

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

## Acknowledgements

We are grateful to Drs Dimitri Kullmann, James Rothman, and Dmitri Rusakov for reading the manuscript and providing critical feedback. This work was supported by National Institute of Health (NIH) grant NS133091 (S.S.K. and K.E.V.); UKRI MRC Project Grant MR/T002786/1 (Y.T. and K.E.V.); UKRI BBSRC/NC3R Project Grant NC/X002233/1 (K.E.V.).

## Author contributions

S.S.K. and K.E.V. conceived the project; D.B. and M.B. carried out the in vitro functional analysis; C.A.N. and Y.T. contributed to the implementation of the computational model; C.A.N. performed all model simulations. S.S.K. and K.E.V. wrote the manuscript. All authors discussed the results and commented on the manuscript.

## Competing interests

The authors declare no competing interests.
