## [Transparent Peer Review file · Nature Communications]

Minimal presynaptic protein machinery governing diverse kinetics of calcium-evoked neurotransmitter release

Corresponding Author: Dr Shyam Krishnakumar

Version 0:

Reviewer comments:

Reviewer #1

(Remarks to the Author)

This study by Bose and colleagues shows that, in an in-vitro biochemical assay, synthetic vesicles containing VAMP2 fuse with membranes containing SNAP-25, Syntaxin-1 and Complexin with different kinetics depending on the proportion of Synaptotagmin-1 to Synaptotagmin-7 that is present. Specifically, they show that, when Synaptotagmin-1 alone is present, vesicles fuse quickly after the addition of calcium (usually within one or two seconds), while the addition of Synaptotagmin-7 delays this process in a concentration-dependent manner. It further provides some evidence that these two Synaptotagmin proteins compete to bind the same SNAREs. Finally, it shows that a “release of inhibition” computational model can explain their observations.

The molecular mechanisms that mediate vesicular fusion at synapses have been broadly studied for decades. In particular, a large body of work at the physiology, molecular biology and biochemistry levels showed that Synaptotagmin-1 and -7 (for reviews, see for example PMID 24183019 or 32278209) have different roles mediating the timing of vesicular fusion. Although the study by Bose and colleagues mostly supports those previous conclusions, I consider that the evidence that they present using a minimal fusion assay provides a valuable addition. Hence, I consider that this study deserves to be published. However, there are major concerns that must be addressed:

1) Validity of the fusion assay. The fusion assay is not fully convincing based on the current examples and details provided by the authors. In particular, the following points need clarification:

-In Figure 1c, why are the “None” and “Syt7” representative examples so different from one another? Why are there no examples from the other conditions?

-How are individual vesicles identified? From the examples provided, one can only see amorphous blobs of fluorescence.

-Why is there so much background? On occasions, it seems impossible to distinguish vesicles from the surrounding background (for example, the bottom circled vesicle corresponding to the “none” example in Figure 1c).

2) Claims of the study. There are multiple claims of synchronous release, asynchronous release, release probability and plasticity. None of these modes of synaptic vesicle fusion is tested in this study. In fact, this appears to be technically not feasible in their system. For example, the temporal resolution of the fusion assay (seconds) is orders of magnitude below what is necessary to categorize fusion as synchronous or asynchronous (milliseconds). Hence, all the claims relative to synchronous, asynchronous, release probability and plasticity have to be removed from the manuscript (title, abstract and main text). The writing should remain factual and describe the true findings of the study: the fusion of vesicles in an in-vitro system and probable competition to bind SNARE machineries. This may require major rewriting. Of course, the authors can speculate how their observation may relate to different modes of fusion at a synapse in the context of a discussion of results. For this, a dedicated discussion section, which is currently not included, would be useful.

3) Level of detail. Relevant information to understand the rationale, bases or conclusion of the experiments is often omitted. This includes the following:

-In figure 2, why are low copy numbers of VAMP2 used? What happens when the experiment is done in close-to-physiological copy numbers of VAMP2?

-The SytQ mutant needs to be properly introduced

-I believe that the conclusion "Syt1 and Syt7 cooperate to establish the fusion clamp at rest" is not accurate. The authors just show that Synaptotagmin-7 can compensate for the lack of Synaptotagmin-1, which does not indicate cooperation. In fact, their experiment suggesting competition to bind to the same SNAREs (figure 3c) indicates the opposite to cooperation. The caption and conclusion of the section should be fixed.

-Figure 3 shows competition between Synaptotagmin proteins in their system. They may want to broaden the interpretation. How does the concentration of the proteins used here relate to the estimated physiological concentrations? At equal molar ratios Syt1:Syt7, > 80% of SNAREs seems to be bound to Syt1. Would this be the case in a synapse?

4) Statistics and experimental design.

-The choice of statistical tests is not correct. To compare multiple groups in figures 1b, 2b, 3a, 3b the authors used t-tests. Tests suitable for multiple groups (ANOVA or Kruskal-Wallis, depending on whether the assumptions of normality and homogeneity of variances are met) have to be used.

-The number of replicates seems intuitively low (four or five replicates). Are there limitations that prevent the authors from having a higher number of replicates?

Reviewer #2

(Remarks to the Author)

Synaptotagmin 7 (Syt7) is a key regulator of neurotransmitter release, but many aspects of its function have been mysterious. The authors take a new approach to teasing out some of these issues by employing a biochemically defined system they've developed, in conjunction with modeling. They use this system to probe the mechanistic origins of two key aspects of Syt7 function, which both regulates asynchronous release and promotes short-term facilitation. Both of these are recapitulated in 'minimalist' in vitro experiments with syntaxin/SNAP25/Syt7 on suspended bilayers, VAMP2/Syt1 on liposomes, and complexin in solution; see Fig. 1a. The data are quite clean, and are consistent with the idea that Syt1 and Syt7 compete for binding to the same SNARE complexes, both capable of functioning as fusion clamps but with intrinsically distinct rates of release. Modeling reproduces these results. The data are also consistent with the hypothesis that Syt7-mediated facilitation involves the pre-weakening of SNARE-Syt7 complexes at low calcium concentrations. Overall I think this work is interesting, clarifying, and of sufficiently broad interest to justify serious consideration for publication in Nature Communications.

1. Unless I missed it, the authors did not justify their placing Syt7 on the pre-synaptic membrane rather than the vesicle. I wonder if they think their results might actually shed light on this still somewhat contentious question?

2. It is surprising to me that the model predictions are largely independent of the model parameters. The most surprising, I'd say, is that the predictions don't appear to depend much if at all on the stoichiometry of the SNARE-synaptotagmin interaction (1:1 versus 1:2). This merits discussion/explanation.

3. Fig. 5 would be strengthened by repeating the Syt1wt/Syt7wt experiment and modeling with various initial calcium concentrations and comparing the two.

Reviewer #3

(Remarks to the Author)

In this manuscript, Bose et al. used a reconstitution approach to study different modes (synchronous and asynchronous) of Ca²⁺-triggered SNARE-mediated membrane fusion in vitro. This is potentially interesting work that might help the field understand the molecular basis for synchronous and asynchronous neurotransmitter release. The authors studied two Ca²⁺ sensors, Syt1 and Syt7, and showed that both clamped SNARE mediated fusion. Upon receiving a Ca²⁺ signal, Syt1 induced rapid vesicle fusion while Syt7 favored slower fusion in their reconstituted system. The authors conclude that control of different modes of Ca²⁺ triggered vesicle fusion occurs through the competitive binding of Syt1 and 7 to SNARE proteins. The assay system is potentially very powerful, but there are a series of issues concerning the biological relevance of the major findings, as well as a number of concerns regarding the experiments and interpretation, as detailed below in no specific order.

1) Some of the claims made by the authors do not seem to be supported by the literature. This is a serious issue that impacts the experimental findings in the current study. A few important examples are:

a) While complexin appears to function as a fusion clamp in *Drosophila* and *C. elegans*, this does not seem to be the case in mammalian neurons. The preponderance of data, from mammalian neurons, strongly support the conclusion that complexin is not a fusion clamp (e.g., this study from the Brose lab, DOI: 10.1016/j.celrep.2019.02.030, among many others).

b) It was recently shown, by the Watanabe lab, that Syt7 does not drive asynchronous release in hippocampal neurons; rather, syt7 regulates docking dynamics, providing vesicles for asynchronous release triggered by Doc2 (DOI: 10.1016/j.celrep.2019.02.030). These findings potentially explain the observation that asynchronous release, in response to a single stimulus, is unaffected in most Syt7 KO synapses. At synapses, the syt7 KO phenotype tends to emerge when more than one stimulus is applied. In contrast to neuron-based experiments, in the *in vitro* system described in the current study, Syt7 had no effect on docking.

c) Syt7 KO neurons do not show changes in spontaneous release rates, arguing against a physiologically relevant clamping function.

These are a few examples of how the authors of the current study have reconstituted what could be argued as biologically irrelevant functions for syt7 and complexin in mammalian neurons.

2) The findings that Ca²⁺ ligand mutations in the C2B domain of syt1 have been reported to have potent dominant negative activity (starting with this study: DOI: 10.1038/nature00846) do not seem to have been taken into consideration RE the experimental findings in the current manuscript.

3) This reviewer had difficulty following the argument regarding the “release of inhibition model”. There is strong evidence that syt1 plays a positive role in the regulation of fusion. For example, it must penetrate membranes to trigger exocytosis, and it has been shown - *in vitro* - to directly facilitate the assembly of trans-SNARE complexes in response to Ca²⁺ (e.g. doi: 10.1038/nsmb.1463). Clearly, Ca²⁺ promotes the interaction of syt1 with t-SNAREs. Together, these findings would seem to be at odds with the authors claim that “The fast removal of the Syt1 clamp may be attributed to the rapid dissociation of Syt1 from the SNARE complex upon Ca²⁺-triggered membrane insertion of its C2B domain, as demonstrated in biochemical and structural studies^{29,30}”. The referee acknowledges the citation of two *in vitro* studies suggesting that inclusion of PIP2 in biochemical assays results in dissociation of Syt1 from SNARE proteins, but this idea remains the subject of debate.

4) Another source of confusion: the authors frequently mention CPX works synergistically with Syt1 and 7 to mediate clamping (again, CPX is unlikely to function as a clamp in mammalian neurons, as discussed above), while in some experiments they included CPX and in others they did not (at least they did not point out whether CPX was present). For example, Figure 2 is from a CPX-free condition, but in Figure 3AB, it is not specified whether CPX was present or not. This needs to be clarified.

5) In Figure 4 the authors claimed that “Syt1 and Syt7 exhibit comparable activation patterns, as their rates of Ca²⁺ binding and membrane association are similar under saturating [Ca²⁺]”. This was the rationale for the author’s conclusion that the Ca²⁺ binding and membrane association rates of Syt7 are not the rate-limiting step, but rather the delay between the membrane insertion step and removal of the clamp constitutes the rate limiting step. However, the authors did not provide any data or citations to support these claims.

6) In the title of Figure 5, the authors claim that “Syt7 enhanced synchronized fusion under elevated [Ca²⁺] basal conditions”, but the data show that under elevated basal [Ca²⁺], synchronized release was decreased in the presence of Syt7 (when compared to Syt1 only).

7) Further RE Figure 5, the only place where basal Ca²⁺ made any difference was when Syt (WT or mutant) was co-reconstituted with Syt7, but this cannot be interpreted because a crucial control, Syt7 alone, was not included. Without this crucial control, it is also difficult to interpret the lack of a difference in the data in samples bearing WT or mutant Syt1.

8) While the main text indicates the authors are using WT recombinant Syt7, it is clear from the Methods that the WT protein was not used in this study. Namely, the authors state: “In addition, we created and utilized a full-length Syt7 clone, which contained rat Syt7 residues 17-403 attached to the Syt1 transmembrane domain (TMD) with a flexible 16 residue GSGS linker and a N-terminal SUMO tag (SUMO-Syt1TMD-Syt7). Note: We included Syt1TMD to the N-terminus of full-length Syt7 to enhance the protein’s reconstitution efficiency in the membrane, while the flexible linker ensured the proper orientation of the Syt7 C2AB domain.” This reviewer is unable to decipher the structure of the Syt7 chimera that was used, because residue 17 seems to be the start of the transmembrane domain of Syt7, so such a construct would contain the Syt7 transmembrane domain, yet the authors state they added the transmembrane domain of Syt1. So, the construct has two transmembrane domains? The discussion of the addition of a linker was also confusing; Syt7 already has a linker between its transmembrane and C2A domain.

Minor:

1) It would be helpful to define each state (So, S1 etc.) in Figure 4B.

2) In Supplemental Figure 1, please label the molecular weight markers.

Version 1:

Reviewer comments:

Reviewer #1

(Remarks to the Author)

The authors have improved the manuscript. However, my previous concern that the authors overinterpret their results has not been addressed. As I stated earlier, the current methodology (biophysical fusion assays and modeling accounting for changes in Ca²⁺ concentration) does not allow drawing any conclusion on plasticity. Claims about this phenomenon should disappear from the title, abstract, and introduction. The authors can connect their study to previous literature that studies plasticity in the discussion.

Other than that, I have no further remarks.

Reviewer #2

(Remarks to the Author)

The authors have addressed my concerns.

Reviewer #3

(Remarks to the Author)

The manuscript has been improved, and the authors have clarified some points, but a number of points still need to be added or addressed, as listed below, and in no specific order. The assay continues to be innovative and potentially powerful, and while no new experiments are required, major revisions to the text are required to better balance the findings and claims in this study with what is known about the function of complexin and syt7 in mammalian neurons, and the biochemical properties of syt7.

a. Although the role of complexin was initially controversial - whether it clamps or plays a positive role in fusion - it is now much more evident that it plays a positive role in fusion in mammalian neurons. The authors should explain this clearly in the text. Removing complexin from cultured neurons does not lead to enhanced fusion.

b. Another point about complexin: since Fig 2 did not include complexin at all, it is misleading to write in the title that "Syt7 acts along with Syt1 and CPX to establish the fusion clamp" and similar claims in the manuscript. Furthermore, as the authors pointed out in the rebuttal letter, the "impact of Syt7 on Ca²⁺-evoked fusion is not affected by absence or presence of CPX" also suggests that Syt7 does not work along with complexin.

c. With respect to point 12 in the rebuttal, Syt1 and Syt7 do not exhibit the same binding affinity and membrane binding properties. In fact, Syt1 is a low affinity, fast sensor, whereas syt7 is a high affinity, slow sensor for Ca²⁺. See, for example: [10.1093/emboj/21.3.270](https://doi.org/10.1093/emboj/21.3.270), [10.1091/mbc.e05-04-0277](https://doi.org/10.1091/mbc.e05-04-0277), [10.1073/pnas.0500941102](https://doi.org/10.1073/pnas.0500941102), [10.1091/mbc.E17-11-0623](https://doi.org/10.1091/mbc.E17-11-0623), [10.1016/j.conb.2020.02.006](https://doi.org/10.1016/j.conb.2020.02.006).

d. As per their biophysical and biochemical properties mentioned above, Syt1 and Syt7 do not have comparable activation patterns. Given that Syt1 DA can act as a dominant negative - and without a syt7 alone control - it cannot be concluded that syt7 unclamping isn't the rate-limiting step.

e. In Fig 5, the revised title is still misleading. To this reviewer, this figure shows basal Ca²⁺ increases the Ca²⁺ synchronized fusion with the presence of both Syt7 and Syt1.

f. In the abstract, the conclusions regarding the findings in the study appear to be overstated, e.g. "...a direct demonstration that a small set of proteins is sufficient to account for how nerve terminals adapt and regulate the Ca²⁺-evoked neurotransmitter exocytosis process...". As discussed in detail in the prior review, the findings in the reconstituted system used throughout this study are sometimes at odds with the apparent roles of these proteins in synapses. This is not particularly surprising, as reconstituted systems and a ground-up approach contains only a subset of proteins. What is needed here is to be open and transparent when the reconstituted systems fail to recapitulate the biology. The approaches used in this study are innovative and powerful, but all reconstitution studies have limitations and it incumbent on the authors to acknowledge these. The author do acknowledge, in their rebuttal, the points raised: neither complexin nor Syt7 are physiologically relevant fusion clamps in most mammalian synapses (also indicated in point (a) above).

g. Since the Syt7 chimera in this manuscript contains two TMDs, is there any supplemental data showing the Syt7 chimera works similarly to WT Syt7? The authors still refer to this chimeric Syt7 as WT – this should be corrected as it is not the wild type protein.

Version 2:

Reviewer comments:

Reviewer #1

(Remarks to the Author)

No further comments.

Reviewer #3

(Remarks to the Author)

The authors have addressed several issues, but there are a couple of matters that still need revision, as follows:

1) The authors have now distinguished between the biophysical properties of syt1 and syt7. However, the explanation of membrane dissociation rates of syt1 and syt7 has a discrepancy between the stated facts and k_{out} values. Specifically, syt7 has a slower membrane dissociation rate ($k_{diss} \sim 19.7 \text{ s}^{-1}$) than the faster rate of syt1 ($k_{diss} \sim 378 \text{ s}^{-1}$). In modeling membrane insertion of the loops of syts (Supplementary Fig. 5), the authors have used incorrect values of k_{out} for syt1 and syt7; in fact, the values they used are almost interchanged between the two isoforms (with syt1 being slow and syt7 being fast).

2) As suggested by Reviewer #1, 'facilitation' should not be used to describe in vitro findings (see Fig. 5 legend).

Version 3:

Reviewer comments:

Reviewer #3

(Remarks to the Author)

When I submitted my last review I subsequently realized I was incorrect when I called the rate constant values, used by the authors, into question. I then wrote to Nature Comm. to withdraw that criticism but it seems that message did not make it to the authors. I apologize for the time the authors spent to address a concern that was not valid. The only other concern was minor, and it was addressed. I have no further issues and support publication. These are difficult experiments, and while I questioned some of the biological relevance of some of the observations that were made, I am enthusiastic about the efforts of these authors to reconstitute docking and fusion in such an elegant manner.

Manuscript Number: NCOMMS-24-17720-T

We are grateful for the thoughtful analysis by the reviewers, and it has been very helpful in revising the manuscript. Below we respond to individual reviewer comments (in blue) in a point-by-point manner.

Reviewer 1

1) Validity of the fusion assay. The fusion assay is not fully convincing based on the current examples and details provided by the authors. In particular, the following points need clarification: In Figure 1c, why are the “None” and “Syt7” representative examples so different from one another? Why are there no examples from the other conditions? How are individual vesicles identified? From the examples provided, one can only see amorphous blobs of fluorescence. Why is there so much background? On occasions, it seems impossible to distinguish vesicles from the surrounding background (for example, the bottom circled vesicle corresponding to the “none” example in Figure 1c).

We create the suspended bilayer by rupturing giant unilamellar vesicles (GUVs) on Si/SiO₂ chips that have a regular array of 5 μm diameter holes. We track individual vesicles only within the suspended bilayer areas using fluorescence signal of the ATTO 647 lipid marker. The background seen in the original version of Figure 1C is caused by autofluorescence from silicon at 647 nm excitation and varies from chip to chip. However, the fluorescence background in this channel is low within the suspended bilayer (see image below), making it easy to identify docking and the fate of individual vesicles. We have revised Figure 1C to focus only on the 5 μm suspended bilayer areas.

2) Claims of the study. There are multiple claims of synchronous release, asynchronous release, release probability and plasticity. None of these modes of synaptic vesicle fusion is tested in this study. In fact, this appears to be technically not feasible in their system. For example, the temporal resolution of the fusion assay (seconds) is orders of magnitude below what is necessary to categorize fusion as synchronous or asynchronous (milliseconds). Hence, all the claims relative to synchronous, asynchronous, release probability and plasticity have to be removed from the manuscript (title, abstract and main text). The writing should remain factual and describe the true

findings of the study: the fusion of vesicles in an in-vitro system and probable competition to bind SNARE machineries. This may require major rewriting. Of course, the authors can speculate how their observation may relate to different modes of fusion at a synapse in the context of a discussion of results. For this, a dedicated discussion section, which is currently not included, would be useful.

We agree with the reviewer and follow their advice. Indeed, the temporal resolution of our fusion assay is limited by the shape of the Ca^{2+} signal and due to technical limitations, we cannot replicate the rapid, millisecond-scale Ca^{2+} transients evoked by action potentials at the presynaptic active zone. Nonetheless, we observe that the rate of fast Syt1-mediated fusion closely mirrors the rate of $[\text{Ca}^{2+}]$ increase at the lipid bilayer. Furthermore, the computational implementation of the 'release of inhibition' model allows us to correlate our experimental findings to the millisecond kinetics of vesicular fusion in response to fast $[\text{Ca}^{2+}]$ transients observed in live synapses¹. Hence, we believe that the reconstituted fusion assays replicate the functionality of the component proteins in living synapses and as such, Syt1 and Syt7 are likely sufficient to describe the synchronous and asynchronous components of AP-evoked neurotransmitter release in neuronal synapses.

We have now included an expanded discussion on how our results may relate to synchronous and asynchronous components of release in synapses. Further, we have revised the title to read: "*Minimal presynaptic protein machinery governing kinetics and plasticity of calcium-evoked neurotransmitter release*".

3) Level of detail. Relevant information to understand the rationale, bases or conclusion of the experiments is often omitted. This includes the following:

a) In figure 2, why are low copy numbers of VAMP2 used? What happens when the experiment is done in close-to-physiological copy numbers of VAMP2?

We used the low VAMP2 condition, combined with Syt1^Q mutation, to isolate and study the role of Syt7 in establishing the fusion clamp. We were unable to simply remove Syt1 and/or CPX under physiologically-relevant conditions due to practical constraints: omitting CPX led to increased spontaneous fusion, while excluding Syt1 significantly reduced the number of docked vesicles. Consequently, we developed this specific reconstitution conditions to probe Syt7's role as a fusion clamp.

We re-worded this section in the revised manuscript as follows: *The clamping efficiency of docked vesicles was similar in the absence or presence of Syt7, with approximately 90% of docked vesicles stably clamped under both conditions (Supplementary Figure 2). Therefore, it remained unclear whether Syt7 also contributes to the fusion clamp (in addition to Syt1 and CPX) under these conditions. Simple removal of Syt1 and/or CPX from the reaction mixture was not feasible, as omitting CPX potentiated spontaneous fusion, while leaving out Syt1 significantly reduced the number of docked vesicles, precluding any meaningful analysis. Hence, we developed reconstitution conditions specifically tailored to investigate the role of Syt7 as a fusion clamp.*

In previous work, we demonstrated that CPX could be omitted under low VAMP2 copy number conditions (i.e. vesicles containing ~13 copies of VAMP2 and ~22 copies of Syt1), as Syt1 alone could produce stably clamped Ca^{2+} -sensitive vesicles^{2,3}. We further showed that disrupting the

Syt1-SNARE interaction at the 'primary' interface using the well-established mutations in the Syt1 C2B domain (R281A, E295A, Y338W, R398A, R399A; referred to as Syt1^Q)^{4,5} abrogates the Syt1 fusion clamp³ without affecting vesicle docking. Given this background, we investigated Syt7's impact on the fusion clamp by utilizing the Syt1^Q mutant in the CPX-free, low VAMP2 condition (Figure 2).

b) The Syt1^Q mutant needs to be properly introduced.

We have included the following clarification in the revised manuscript: *... "disrupting the interaction between Syt1 and t-SNARE at the 'primary' interface using the structurally-designed mutations in the Syt1 C2B domain (R281A, E295A, Y338W, R398A, R399A; referred to as Syt1^Q)^{4,5} abolished the Syt1 fusion clamp^{2,3}"*

c) I believe that the conclusion "Syt1 and Syt7 cooperate to establish the fusion clamp at rest" is not accurate. The authors just show that Synaptotagmin-7 can compensate for the lack of Synaptotagmin-1, which does not indicate cooperation. In fact, their experiment suggesting competition to bind to the same SNAREs (figure 3c) indicates the opposite to cooperation. The caption and conclusion of the section should be fixed.

Our data indicate that Syt1, Syt7, and CPX all play a role in establishing the fusion clamp. Since Syt1 and Syt7 compete to bind the same SNARE complex, we infer that an RRP vesicle likely contains a mixture of Syt1/CPX and Syt7/CPX clamped SNAREpins, depending on the local abundance of Syt1 and Syt7. This is accurately reflected in our conclusion that *... 'under physiologically relevant conditions, Syt7 acts in concert with Syt1 and CPX to arrest SNARE assembly and produce stably docked vesicles at rest'*. However, we agree that it is misleading to state that Syt1 and Syt7 cooperate to establish the fusion clamp. Therefore, we have revised the title to *'Syt7 acts along with Syt1 and CPX to establish the fusion clamp'*.

D) Figure 3 shows competition between Synaptotagmin proteins in their system. They may want to broaden the interpretation. How does the concentration of the proteins used here relate to the estimated physiological concentrations? At equal molar ratios Syt1:Syt7, > 80% of SNAREs seems to be bound to Syt1. Would this be the case in a synapse?

We agree. We have included the following paragraph in the Discussion section: *"Moreover, we found that Syt1 and Syt7 compete to bind the same SNARE complexes. Consequently, the relative abundance of these Ca²⁺ sensors shaped the kinetics and plasticity of Ca²⁺-evoked vesicular fusion in our assay. This observation offers a mechanistic explanation for physiological data demonstrating that the relative expression levels of Syt1 and Syt7 regulate both the kinetics and plasticity of neurotransmitter release in neuronal synapses^{6,7}. The number of Syt1 per vesicle is tightly controlled (~15-20 copies per SV) and local Syt1 concentration under a docked vesicle is estimated to be in the range of 5 – 10 mM^{8,9}. Syt1 interacts with SNARE complex at the primary interface with an affinity of ~10 μM, suggesting a near-complete saturation of Syt1-SNARE binding under physiological conditions. However, the precise concentration of Syt7 is unknown and varies across synapses. Consequently, the number of Syt1- or Syt7-bound SNAREpins under a given RRP vesicle is likely determined by local abundance of Syt7. This implies that controlling the local abundance of Syt7 molecules could be straightforward mechanism by which synapses dynamically adjust the strength and efficacy of synaptic transmission during sustained activity".*

4) Statistics and experimental design.

a) The choice of statistical tests is not correct. To compare multiple groups in figures 1b, 2b, 3a, 3b the authors used t-tests. Tests suitable for multiple groups (ANOVA or Kruskal-Wallis, depending on whether the assumptions of normality and homogeneity of variances are met) have to be used.

As recommended, we have carried out the statistical analysis using one-way ANOVA, followed by post-hoc Tukey HSD between specific groups. The statistical analysis results are shown in Supplementary Tables 1-5. We wish to note that the choice of statistical test does not alter our conclusions.

b) The number of replicates seems intuitively low (four or five replicates). Are there limitations that prevent the authors from having a higher number of replicates?

In this biochemically-defined system, all key components such as the identity and density of proteins, composition of buffers, lipid membranes, and Ca^{2+} signal are independently and rigorously controlled. The statistical unit of replication are recordings from a single reconstitution experiment in which we typically monitor the docking and fusion of ~40-60 vesicles. The standard deviation in these experiments is in the range of 5-10%. Therefore, we typically carry out 3 - 5 independent recordings (~150-200 vesicles in total) per condition for reliable comparison of fusion probabilities.

Reviewer 2

5. Unless I missed it, the authors did not justify their placing Syt7 on the pre-synaptic membrane rather than the vesicle. I wonder if they think their results might actually shed light on this still somewhat contentious question?

To the best of our knowledge, in central synapses Syt7 predominantly localizes to the pre-synaptic plasma membrane^{10,11} and is not found in synaptic vesicle isolations¹². Hence, we chose to include Syt7 along with the t-SNARE in the suspended bilayer. We have included the following statement in the Results section (instead of the Method section in the original manuscript) of the revised manuscript: *“We incorporated Syt7 in the suspended lipid membrane reflecting its predominant localization in the presynaptic membrane in central synapses.”*

6. It is surprising to me that the model predictions are largely independent of the model parameters. The most surprising, I'd say, is that the predictions don't appear to depend much if at all on the stoichiometry of the SNARE-synaptotagmin interaction (1:1 versus 1:2). This merits discussion/explanation.

As depicted in Supplementary Figure 5, increasing the number of SNAREpins, or reducing the clamp strength in a given RRP vesicle improves the fusion rate, but only at low $[\text{Ca}^{2+}]$. There is no effect observed at higher saturating $[\text{Ca}^{2+}]$ levels, similar to that used in our assay. Importantly, these modifications do not alter the overarching conclusion that mechanisms of clamp removal are distinct for Syt1 and Syt7 and that the differing rates of clamp removal – rapid for Syt1 and slower for Syt7 – are key factors determining the Ca^{2+} triggered vesicular fusion kinetics.

We have included the following clarification in the revised manuscript: *“As expected, increasing the number of SNAREpins, or using a single clamp instead of a dual clamp accelerated fusion*

rates at low $[Ca^{2+}]$. However, at saturating $[Ca^{2+}]$, the models' outputs closely aligned with those of the default dual-clamp model with six SNAREpins (Supplementary Figure 5).

7. Fig. 5 would be strengthened by repeating the Syt1wt/Syt7wt experiment and modeling with various initial calcium concentrations and comparing the two.

We indeed examined the effect of varying the $[Ca^{2+}]_{\text{basal}}$ on facilitating Ca^{2+} -evoked release in both modeling and experiments. In the in vitro assay we observed that $[Ca^{2+}]_{\text{basal}}$ of 1 μM significantly increased spontaneous fusion of docked vesicles, making analysis impractical. Therefore, we selected 0.5 μM as the optimal initial $[Ca^{2+}]$ for the in vitro assay. We note that in central synapses during short high-frequency burst of action potentials $[Ca^{2+}]_{\text{residual}}$ that mediates short-term facilitation typically remains in sub-micromolar range.

We have included the following text in the revised manuscript: *"In our in vitro functional assay we only tested 0.5 μM $[Ca^{2+}]_{\text{basal}}$, as higher $[Ca^{2+}]_{\text{basal}}$ significantly increased spontaneous fusion of docked vesicles, preventing meaningful analysis. However, as expected, the degree of synchronization predicted by the model correlated with the concentration of Ca^{2+} utilized for pre-activation (Figure 5B)."*

Reviewer 3

8) Some of the claims made by the authors do not seem to be supported by the literature. This is a serious issue that impacts the experimental findings in the current study. A few important examples are:

a) While complexin appears to function as a fusion clamp in *Drosophila* and *C. elegans*, this does not seem to be the case in mammalian neurons. The preponderance of data, from mammalian neurons, strongly support the conclusion that complexin is not a fusion clamp (e.g., this study from the Brose lab, DOI: 10.1016/j.celrep.2019.02.030, among many others).

While it is true that most studies of cultured hippocampal neurons have not observed enhanced spontaneous fusion when mammalian CPX1/2/3 are removed¹³, it is worth noting that under these conditions CPX removal abates both spontaneous and evoked neurotransmitter release without changing the number of docked vesicles¹³. This suggests that acute CPX loss likely affects the late-stage vesicle priming process, and it is possible this 'loss-of-fusion' phenotype occludes CPX role in regulating spontaneous fusion events. Indeed, a recent study examining accessory helix mutations did report a 2-fold increase in spontaneous fusion using a mutant mCpx1 in the context of the mammalian CPX 1/2/3 triple KO, indicating that there still could be inhibitory function that is normally masked by a more pronounced positive function¹⁴. Several papers examining mammalian CPX 3/4 in the retina uncovered a strong inhibitory function of these isoforms^{15,16}. And at the calyx of Held, mammalian Cpx1 was observed to stabilize newly docked SVs and prevent premature spontaneous fusion¹⁷. Furthermore, there is compelling evidence from in vitro reconstitution studies, including our assay, that mammalian CPX is an integral part of the clamping mechanism^{3,18}. CPX plays a critical role, alongside Synaptotagmin isoforms, in clamping SNARE mediated spontaneous fusion and producing stably docked vesicles. We would also like to point out that the impact of Syt7 on Ca^{2+} -evoked fusion is not affected by absence or presence of CPX (see Figure 1 vs. Figure 2).

b) It was recently shown, by the Watanabe lab, that Syt7 does not drive asynchronous release in hippocampal neurons; rather, syt7 regulates docking dynamics, providing vesicles for asynchronous release triggered by Doc2 (DOI: 10.1016/j.celrep.2019.02.030). These findings potentially explain the observation that asynchronous release, in response to a single stimulus, is unaffected in most Syt7 KO synapses. At synapses, the syt7 KO phenotype tends to emerge when more than one stimulus is applied. In contrast to neuron-based experiments, in the in vitro system described in the current study, Syt7 had no effect on docking.

The identity of the Ca^{2+} sensor involved in mediating asynchronous release remains a topic of debate. Evidence suggests that Syt7 contributes to asynchronous release in various synapses, particularly following sustained activity^{6,19-23}. Conversely, other studies propose that other Ca^{2+} -sensors such as Doc2b might trigger asynchronous release, while Syt7 is implicated in the vesicle docking process^{24,25}. Due to the overlapping functions and variable expression levels across synapses, elucidating the specific roles of individual Ca^{2+} sensors in live synapses remains challenging. To address this issue, we employed a biochemically-defined fusion assay with a minimal set of defined components.

Our findings demonstrate that Syt7 possesses an intrinsic capability to induce delayed fusion of docked vesicles in response to a Ca^{2+} signal, suggesting its potential role as a Ca^{2+} sensor for the asynchronous release component. Given the nature of the Ca^{2+} signal in our in vitro assay, our results are more pertinent to delayed release during bursts rather than single action potentials. Further investigations employing controlled Ca^{2+} transients are necessary to confirm whether Syt7 can also facilitate delayed release in response to a single AP stimulus. As mentioned earlier, we have adjusted the terminology and expanded our discussion to elucidate how our findings contribute to understanding the synchronous and asynchronous components of synaptic vesicle release observed in neurons.

We have included the following statement in the Introduction (in addition to the Discussion section) to highlight this issue: “*However, Syt7’s role in asynchronous release remains a topic of debate as other Ca^{2+} -release sensors (e.g. Syt1 or Doc2A) have also been implicated in mediating the asynchronous release component^{24,25}.*”

c) Syt7 KO neurons do not show changes in spontaneous release rates, arguing against a physiologically relevant clamping function.

Deletion or overexpression of Syt7 under WT condition does not alter frequency of spontaneous release²⁶. However, Syt7 overexpression reverses the increased mini frequency in Syt1^{KO} neurons arguing that Syt7 can substitute for Syt1 in clamping mini release²⁶. Syt7 clamping function may not be apparent under physiological conditions because Syt7 may not be expressed at sufficiently high levels, especially within presynaptic terminals. Indeed, we find Syt7 clamping function correlates to the amount of Syt7 included in the assay. We have included the following statement in the revised manuscript: “*This indicates that Syt7 can independently establish a Ca^{2+} -sensitive fusion clamp. It further suggests that, under physiologically relevant conditions, Syt7 acts in concert with Syt1 and CPX to arrest SNARE assembly and produce stably, docked vesicles at rest. The deletion of Syt7 under physiological condition does not alter frequency of spontaneous release²⁶. However, Syt7 overexpression reverses the increased mini frequency in Syt1 knockout neurons²⁶. This suggests that Syt7 can substitute for Syt1 in clamping mini release, but its*

clamping role may not be apparent under normal physiological conditions due to low expression levels of Syt7 in presynaptic terminals.”

d) These are a few examples of how the authors of the current study have reconstituted what could be argued as biologically irrelevant functions for syt7 and complexin in mammalian neurons.

We have reconstituted genetically-verified components of SV exocytic machinery under physiologically relevant conditions and geometry. The distinct effects of different truncation and targeted mutations in these proteins, namely SNARE, Syt1, CPX, Munc18 and Munc13, match with data obtained from other reductionist or physiological systems^{2,3,27-29} forcefully arguing for the physiological relevance of results obtained from our in vitro reconstituted assay.

9) The findings that Ca²⁺ ligand mutations in the C2B domain of syt1 have been reported to have potent dominant negative activity (starting with this study: DOI: 10.1038/nature00846) do not seem to have been taken into consideration RE the experimental findings in the current manuscript.

We observe that the Syt1^{DA} blocks vesicle fusion; however, the extent of its dominant negative effect correlates with the amount of Syt7 present. Given that the Syt1^{DA} and Syt7 compete to bind the same SNARE complex, it is likely that Syt7 replaces Syt1^{DA} on the SNAREs, especially at high concentrations. This might not be apparent in physiological conditions because Syt7 may not be expressed at sufficiently high levels, especially within presynaptic terminals. It is worth noting that we tested the effect of varying Syt7 concentrations as the precise concentration of Syt7 in a synapse is not known.

10) This reviewer had difficulty following the argument regarding the “release of inhibition model”. There is strong evidence that syt1 plays a positive role in the regulation of fusion. For example, it must penetrate membranes to trigger exocytosis, and it has been shown - in vitro - to directly facilitate the assembly of trans-SNARE complexes in response to Ca²⁺ (e.g. doi: 10.1038/nsmb.1463). Clearly, Ca²⁺ promotes the interaction of syt1 with t-SNAREs. Together, these findings would seem to be at odds with the authors claim that “The fast removal of the Syt1 clamp may be attributed to the rapid dissociation of Syt1 from the SNARE complex upon Ca²⁺-triggered membrane insertion of its C2B domain, as demonstrated in biochemical and structural studies^{29,30}”. The referee acknowledges the citation of two in vitro studies suggesting that inclusion of PIP2 in biochemical assays results in dissociation of Syt1 from SNARE proteins, but this idea remains the subject of debate.

We have previously³⁰ demonstrated that Ca²⁺ increases the affinity of Syt1 to the SNARE complexes, ONLY if calcium is included during the binding process. However, adding Ca²⁺ to Syt1-SNARE complex pre-assembled under Ca²⁺-free conditions does not alter the interaction. This suggests that while Ca²⁺ does affect the Syt1-SNARE interaction, the rate of this reaction is much slower (order of minutes) compared to the time scale of Ca²⁺-triggered membrane insertion of Syt1. Hence, we conclude that Ca²⁺-activation of Syt1 is the more relevant molecular process involved in triggering ultra-fast fusion of docked vesicles. Indeed, using the computational modeling, we have recently demonstrated that ‘release of inhibition’ model can explain the millisecond kinetics and plasticity of neurotransmitter release in response to physiologically-relevant Ca²⁺ transients¹.

Whether Syt1 fully dissociates from the SNARE is indeed unclear. Some FRET analyses have shown that some form of Syt1-SNARE interaction remains even in the presence of Ca^{2+} ^{31,32}. Nonetheless, it is evident from biochemical and structural analysis that the Syt1-SNARE interaction at the primary interface, which is crucial for the fusion clamp, is indeed abolished following Ca^{2+} -activation ^{33,34}. We have revised the manuscript to read *'rapid dissociation of Syt1 from the SNARE complex at the primary interface upon triggered membrane insertion of its C2B domain, as demonstrated in biochemical and structural studies'*

11) Another source of confusion: the authors frequently mention CPX works synergistically with Syt1 and 7 to mediate clamping (again, CPX is unlikely to function as a clamp in mammalian neurons, as discussed above), while in some experiments they included CPX and in others they did not (at least they did not point out whether CPX was present). For example, Figure 2 is from a CPX-free condition, but in Figure 3AB, it is not specified whether CPX was present or not. This needs to be clarified.

We have included 2 μM CPX in all experiments except in Figure 2 which is carried out under low VAMP2 conditions. We have indicated this in the legends of Figures 1, 3, & 5.

12) In Figure 4 the authors claimed that "Syt1 and Syt7 exhibit comparable activation patterns, as their rates of Ca^{2+} binding and membrane association are similar under saturating $[\text{Ca}^{2+}]$ ". This was the rationale for the author's conclusion that the Ca^{2+} binding and membrane association rates of Syt7 are not the rate-limiting step, but rather the delay between the membrane insertion step and removal of the clamp constitutes the rate limiting step. However, the authors did not provide any data or citations to support these claims.

The available biochemical and biophysical data indicate that Syt1 and Syt7 have similar intrinsic Ca^{2+} affinity of 150 μM ³⁵⁻³⁷ and a comparable rate of Ca^{2+} -triggered membrane insertion of the C2 domain ($K_{in} = 150 \text{ ms}^{-1}$)⁸. Hence, we conclude that Syt1 and Syt7 exhibit comparable activation pattern under our experimental conditions. We have clarified this section with relevant references as follows: *"Under our experimental conditions, Syt1 and Syt7 should exhibit comparable activation patterns, as their rates of Ca^{2+} binding and membrane association are similar under saturating 100 μM $[\text{Ca}^{2+}]$ "*^{8,35-37}.

13) In the title of Figure 5, the authors claim that "Syt7 enhanced synchronized fusion under elevated $[\text{Ca}^{2+}]$ basal conditions", but the data show that under elevated basal $[\text{Ca}^{2+}]$, synchronized release was decreased in the presence of Syt7 (when compared to Syt1 only).

We agree that the title is confusing. We have revised the title in both the result and figure legend to read: *Syt7 facilitates Ca^{2+} -synchronized fusion under elevated $[\text{Ca}^{2+}]_{\text{basal}}$ conditions.*

14) Further RE Figure 5, the only place where basal Ca^{2+} made any difference was when Syt (WT or mutant) was co-reconstituted with Syt7, but this cannot be interpreted because a crucial control, Syt7 alone, was not included. Without this crucial control, it is also difficult to interpret the lack of a difference in the data in samples bearing WT or mutant Syt1.

As we stated in the manuscript, we are unable to do the experiments under Syt7 only condition as removal of Syt1 severely reduces vesicle docking and potentiates spontaneous fusion of docked vesicles. The fact that we observe facilitation of Ca^{2+} -synchronized release only with Syt7 and even in the presence of non-functional Syt1^{DA} strongly argues that Syt7 acts as the principal

Ca²⁺-sensor for synaptic facilitation during sustained neuronal activity. In addition, we have revised the title for the results section and figure legend to read: “*Syt7 facilitates Ca²⁺ synchronized fusion under elevated [Ca²⁺]_{basal} conditions*”.

15) While the main text indicates the authors are using WT recombinant Syt7, it is clear from the Methods that the WT protein was not used in this study. Namely, the authors state: “In addition, we created and utilized a full-length Syt7 clone, which contained rat Syt7 residues 17-403 attached to the Syt1 transmembrane domain (TMD) with a flexible 16 residue GSGS linker and a N-terminal SUMO tag (SUMO-Syt1TMD-Syt7). Note: We included Syt1TMD to the N-terminus of full-length Syt7 to enhance the protein’s reconstitution efficiency in the membrane, while the flexible linker ensured the proper orientation of the Syt7 C2AB domain.” This reviewer is unable to decipher the structure of the Syt7 chimera that was used, because residue 17 seems to be the start of the transmembrane domain of Syt7, so such a construct would contain the Syt7 transmembrane domain, yet the authors state they added the transmembrane domain of Syt1. So, the construct has two transmembrane domains? The discussion of the addition of a linker was also confusing; Syt7 already has a linker between its transmembrane and C2A domain.

The Syt7 construct used in this study is as follows: SUMO-Syt1^{TMD}-(GSGS)₄-Syt7^{TMD}-linker-C2A-C2B. We have included the following clarification in the Methods section: “*We included Syt1^{TMD} in addition to the Syt7^{TMD} to enhance the reconstitution efficiency of the Syt7 protein into the membrane, while the flexible GSGS linker ensured the proper orientation of the two TMDs and Syt7 C2AB domains (Supplementary Figure 1B)*”. In addition, include a pictorial description of this construct in the Supplementary Figure 1B, along with the vesicle reconstitution and chymotrypsin accessibility assay data.

16) It would be helpful to define each state (So, S1 etc.) in Figure 4B.

We have included the following description in Figure 4B legend: “*The S₀, S₁, S₂ refers with 0, 1 or 2 Ca²⁺ bound state of the C2 domains, while I₂ refers to membrane inserted state of the Ca²⁺-bound C2 domain. The prefixes ^c and ^u refer to the ‘clamped’ or ‘unclamped’ state respectively*”.

17) In Supplemental Figure 1, please label the molecular weight markers.

The MW markers have been labeled in the revised Supplementary Figure 1.

REFERENCES

- 1 Norman, C. A., Krishnakumar, S. S., Timofeeva, Y. & Volynski, K. E. The release of inhibition model reproduces kinetics and plasticity of neurotransmitter release in central synapses. *Commun Biol* **6**, 1091, doi:10.1038/s42003-023-05445-2 (2023).
- 2 Ramakrishnan, S. *et al.* Synaptotagmin oligomers are necessary and can be sufficient to form a Ca(2+) -sensitive fusion clamp. *FEBS Lett* **593**, 154-162, doi:10.1002/1873-3468.13317 (2019).
- 3 Ramakrishnan, S., Bera, M., Coleman, J., Rothman, J. E. & Krishnakumar, S. S. Synergistic roles of Synaptotagmin-1 and complexin in calcium-regulated neuronal exocytosis. *Elife* **9**, doi:10.7554/eLife.54506 (2020).

- 4 Zhou, Q. *et al.* Architecture of the synaptotagmin-SNARE machinery for neuronal exocytosis. *Nature* **525**, 62-67, doi:10.1038/nature14975 (2015).
- 5 Zhou, Q. *et al.* The primed SNARE-complexin-synaptotagmin complex for neuronal exocytosis. *Nature* **548**, 420-425, doi:10.1038/nature23484 (2017).
- 6 Turecek, J. & Regehr, W. G. Neuronal Regulation of Fast Synaptotagmin Isoforms Controls the Relative Contributions of Synchronous and Asynchronous Release. *Neuron* **101**, 938-949 e934, doi:10.1016/j.neuron.2019.01.013 (2019).
- 7 Weyrer, C., Turecek, J., Harrison, B. & Regehr, W. G. Introduction of synaptotagmin 7 promotes facilitation at the climbing fiber to Purkinje cell synapse. *Cell Rep* **36**, 109719, doi:10.1016/j.celrep.2021.109719 (2021).
- 8 Rothman, J. E., Krishnakumar, S. S., Grushin, K. & Pincet, F. Hypothesis - buttressed rings assemble, clamp, and release SNAREpins for synaptic transmission. *FEBS Lett* **591**, 3459-3480, doi:10.1002/1873-3468.12874 (2017).
- 9 Tagliatti, E. *et al.* Synaptotagmin oligomers clamp and regulate different modes of neurotransmitter release. *Proc Natl Acad Sci U S A*, doi:10.1073/pnas.1920403117 (2020).
- 10 Vevea, J. D. *et al.* Synaptotagmin 7 is targeted to the axonal plasma membrane through gamma-secretase processing to promote synaptic vesicle docking in mouse hippocampal neurons. *Elife* **10**, doi:10.7554/eLife.67261 (2021).
- 11 Sugita, S. *et al.* Synaptotagmin VII as a plasma membrane Ca(2+) sensor in exocytosis. *Neuron* **30**, 459-473, doi:10.1016/s0896-6273(01)00290-2 (2001).
- 12 Takamori, S. *et al.* Molecular anatomy of a trafficking organelle. *Cell* **127**, 831-846, doi:10.1016/j.cell.2006.10.030 (2006).
- 13 Lopez-Murcia, F. J., Reim, K., Jahn, O., Taschenberger, H. & Brose, N. Acute Complexin Knockout Abates Spontaneous and Evoked Transmitter Release. *Cell Rep* **26**, 2521-2530 e2525, doi:10.1016/j.celrep.2019.02.030 (2019).
- 14 Malsam, J. *et al.* Complexin Suppresses Spontaneous Exocytosis by Capturing the Membrane-Proximal Regions of VAMP2 and SNAP25. *Cell Rep* **32**, 107926, doi:10.1016/j.celrep.2020.107926 (2020).
- 15 Vaithianathan, T., Henry, D., Akmentin, W. & Matthews, G. Functional roles of complexin in neurotransmitter release at ribbon synapses of mouse retinal bipolar neurons. *J Neurosci* **35**, 4065-4070, doi:10.1523/JNEUROSCI.2703-14.2015 (2015).
- 16 Vaithianathan, T., Zanazzi, G., Henry, D., Akmentin, W. & Matthews, G. Stabilization of spontaneous neurotransmitter release at ribbon synapses by ribbon-specific subtypes of complexin. *J Neurosci* **33**, 8216-8226, doi:10.1523/JNEUROSCI.1280-12.2013 (2013).
- 17 Chang, S. *et al.* Complexin stabilizes newly primed synaptic vesicles and prevents their premature fusion at the mouse calyx of held synapse. *J Neurosci* **35**, 8272-8290, doi:10.1523/JNEUROSCI.4841-14.2015 (2015).
- 18 Bera, M., Ramakrishnan, S., Coleman, J., Krishnakumar, S. S. & Rothman, J. E. Molecular Determinants of Complexin Clamping and Activation Function. *Elife* **e71938** (2022).
- 19 Huson, V. & Regehr, W. G. Diverse roles of Synaptotagmin-7 in regulating vesicle fusion. *Curr Opin Neurobiol* **63**, 42-52, doi:10.1016/j.conb.2020.02.006 (2020).
- 20 Turecek, J., Jackman, S. L. & Regehr, W. G. Synaptotagmin 7 confers frequency invariance onto specialized depressing synapses. *Nature* **551**, 503-506, doi:10.1038/nature24474 (2017).

- 21 Turecek, J. & Regehr, W. G. Synaptotagmin 7 Mediates Both Facilitation and Asynchronous Release at Granule Cell Synapses. *J Neurosci* **38**, 3240-3251, doi:10.1523/JNEUROSCI.3207-17.2018 (2018).
- 22 Chen, C., Satterfield, R., Young, S. M., Jr. & Jonas, P. Triple Function of Synaptotagmin 7 Ensures Efficiency of High-Frequency Transmission at Central GABAergic Synapses. *Cell Rep* **21**, 2082-2089, doi:10.1016/j.celrep.2017.10.122 (2017).
- 23 Luo, F. & Sudhof, T. C. Synaptotagmin-7-Mediated Asynchronous Release Boosts High-Fidelity Synchronous Transmission at a Central Synapse. *Neuron* **94**, 826-839 e823, doi:10.1016/j.neuron.2017.04.020 (2017).
- 24 Wu, Z. *et al.* Synaptotagmin 7 docks synaptic vesicles to support facilitation and Doc2alpha-triggered asynchronous release. *Elife* **12**, doi:10.7554/eLife.90632 (2024).
- 25 Yao, J., Gaffaney, J. D., Kwon, S. E. & Chapman, E. R. Doc2 is a Ca²⁺ sensor required for asynchronous neurotransmitter release. *Cell* **147**, 666-677, doi:10.1016/j.cell.2011.09.046 (2011).
- 26 Bacaj, T. *et al.* Synaptotagmin-1 and synaptotagmin-7 trigger synchronous and asynchronous phases of neurotransmitter release. *Neuron* **80**, 947-959, doi:10.1016/j.neuron.2013.10.026 (2013).
- 27 Bera, M. *et al.* Synaptophysin chaperones the assembly of 12 SNAREpins under each ready-release vesicle. *Proc Natl Acad Sci U S A* **120**, e2311484120, doi:10.1073/pnas.2311484120 (2023).
- 28 Coleman, J. *et al.* PRRT2 Regulates Synaptic Fusion by Directly Modulating SNARE Complex Assembly. *Cell Rep* **22**, 820-831, doi:10.1016/j.celrep.2017.12.056 (2018).
- 29 Kalyana Sundaram, R. V. *et al.* Roles for diacylglycerol in synaptic vesicle priming and release revealed by complete reconstitution of core protein machinery. *Proc Natl Acad Sci U S A* **120**, e2309516120, doi:10.1073/pnas.2309516120 (2023).
- 30 Krishnakumar, S. S. *et al.* Conformational dynamics of calcium-triggered activation of fusion by synaptotagmin. *Biophys J* **105**, 2507-2516, doi:10.1016/j.bpj.2013.10.029 (2013).
- 31 Krishnakumar, S. S. *et al.* Conformational dynamics of calcium-triggered activation of fusion by synaptotagmin. *Biophys J* **105**, 2507-2516, doi:10.1016/j.bpj.2013.10.029 (2013).
- 32 Wang, S., Li, Y. & Ma, C. Synaptotagmin-1 C2B domain interacts simultaneously with SNAREs and membranes to promote membrane fusion. *eLife* **5**, 1-21, doi:10.7554/eLife.14211 (2016).
- 33 Grushin, K. *et al.* Structural basis for the clamping and Ca⁽²⁺⁾ activation of SNARE-mediated fusion by synaptotagmin. *Nat Commun* **10**, 2413, doi:10.1038/s41467-019-10391-x (2019).
- 34 Voleti, R., Jaczynska, K. & Rizo, J. Ca⁽²⁺⁾-dependent release of synaptotagmin-1 from the SNARE complex on phosphatidylinositol 4,5-bisphosphate-containing membranes. *Elife* **9**, doi:10.7554/eLife.57154 (2020).
- 35 Voleti, R., Tomchick, D. R., Sudhof, T. C. & Rizo, J. Exceptionally tight membrane-binding may explain the key role of the synaptotagmin-7 C2A domain in asynchronous neurotransmitter release. *Proc Natl Acad Sci U S A* **114**, E8518-8527, doi:10.1073/pnas.1710708114 (2017).
- 36 Radhakrishnan, A., Stein, A., Jahn, R. & Fasshauer, D. The Ca²⁺ affinity of synaptotagmin 1 is markedly increased by a specific interaction of its C2B domain with phosphatidylinositol 4,5-bisphosphate. *J Biol Chem* **284**, 25749-25760, doi:10.1074/jbc.M109.042499 (2009).

- 37 Davis, A. F. *et al.* Kinetics of synaptotagmin responses to Ca²⁺ and assembly with the core SNARE complex onto membranes. *Neuron* **24**, 363-376, doi:10.1016/s0896-6273(00)80850-8 (1999).

Manuscript Number: NCOMMS-24-17720-A

We are grateful for the continued editorial and reviewer interest in our manuscript. Below we respond to individual reviewer comments (in blue) in a point-by-point manner.

Reviewer #1

The authors have improved the manuscript. However, my previous concern that the authors overinterpret their results has not been addressed. As I stated earlier, the current methodology (biophysical fusion assays and modeling accounting for changes in Ca^{2+} concentration) does not allow drawing any conclusion on plasticity. Claims about this phenomenon should disappear from the title, abstract, and introduction. The authors can connect their study to previous literature that studies plasticity in the discussion.

As recommended, we have revised the title, abstract, and results section to remove references to 'plasticity' and 'facilitation' in relation to our in vitro findings. We have connected our results to physiological studies in the Discussion section

Reviewer # 2

No comments to address.

Reviewer #3

Although the role of complexin was initially controversial - whether it clamps or plays a positive role in fusion - it is now much more evident that it plays a positive role in fusion in mammalian neurons. The authors should explain this clearly in the text. Removing complexin from cultured neurons does not lead to enhanced fusion.

Our *in vitro* functional analysis demonstrates that under physiologically-relevant conditions, (i.e. vesicles containing ~70 VAMP2), Syt1, Syt7 and CPX all contribute to establishing the fusion clamp. We previously showed that Syt1 and CPX synergistically clamp different groups of SNARE complexes, with both required to produce a stably clamped vesicles³. In the current study, we find that Syt7 can substitute for Syt1 in the fusion clamp, suggesting that that a clamped vesicle likely contains a mixture of Syt1/CPX and Syt7/CPX clamped SNAREpins, depending on the local abundance of Syt1 and Syt7.

We acknowledge that the clamping function of CPX and Syt7 in neuronal synapses is debated, and there is a difference between our in vitro findings and physiological studies in mammalian synapses. To address this, we have revised the manuscript to include the following paragraph in the Discussion section that highlights this issue and provides a balanced view.

“While our data indicates that Syt1, Syt7, and CPX all are involved in establishing the fusion clamp, the precise molecular composition of the fusion clamp on a RRP vesicle remains unknown. In particular, the clamping function of CPX and Syt7 remains a topic of debate. For example, genetic removal of CPX potentiates spontaneous events in invertebrate model systems^{4,5}, but acute removal of CPX in cultured mouse neurons abates both spontaneous and evoked neurotransmitter release⁶ suggesting that CPX is principally a positive regulator of fusion in mammalian synapses. However, a recent study showed that CPX mutants that disrupt the clamping function under in vitro conditions^{7,8} selectively potentiate spontaneous neurotransmitter release, while leaving the evoked release largely untouched⁸. This suggests that the inhibitory

function of CPX might be normally masked by a more pronounced positive function in mammalian synapses. Similarly, in central synapses the genetic deletion of Syt7 does not alter frequency of spontaneous release⁹. However, the overexpression of Syt7 reverses the increased mini frequency observed in Syt1 knockout neurons⁹. This suggests that Syt7 can substitute for Syt1 in clamping mini release, but its clamping role may not be apparent under normal physiological conditions due to low expression levels of Syt7 in presynaptic terminals. Additional research is needed to delineate the precise molecular organization of the pre-fusion 'clamped' state and the mechanisms of Ca²⁺-triggered reversal of the fusion clamp.

Another point about complexin: since Fig 2 did not include complexin at all, it is misleading to write in the title that "Syt7 acts along with Syt1 and CPX to establish the fusion clamp" and similar claims in the manuscript. Furthermore, as the authors pointed out in the rebuttal letter, the "impact of Syt7 on Ca²⁺-evoked fusion is not affected by absence or presence of CPX" also suggests that Syt7 does not work along with complexin.

We have retitled this section to read 'Contribution of Syt7 to the establishment of the fusion clamp.

With respect to point 12 in the rebuttal, Syt1 and Syt7 do not exhibit the same binding affinity and membrane binding properties. In fact, Syt1 is a low affinity, fast sensor, whereas syt7 is a high affinity, slow sensor for Ca²⁺. See, for example: [10.1093/emboj/21.3.270](https://doi.org/10.1093/emboj/21.3.270), [10.1091/mbc.e05-04-0277](https://doi.org/10.1091/mbc.e05-04-0277), [10.1073/pnas.0500941102](https://doi.org/10.1073/pnas.0500941102), [10.1091/mbc.E17-11-0623](https://doi.org/10.1091/mbc.E17-11-0623), [10.1016/j.conb.2020.02.006](https://doi.org/10.1016/j.conb.2020.02.006).

As per their biophysical and biochemical properties mentioned above, Syt1 and Syt7 do not have comparable activation patterns. Given that Syt1 DA can act as a dominant negative - and without a syt7 alone control - it cannot be concluded that syt7 unclamping isn't the rate-limiting step.

We recognize that Syt1 and Syt7 exhibit distinct Ca²⁺/membrane binding affinities, with Syt1 exhibiting a lower affinity ($K_d \sim 10\text{-}20 \mu\text{M}$) and Syt7 showing a higher affinity ($K_d \sim 1\text{-}3 \mu\text{M}$)^{1,2}. The key difference in the between Syt1 and Syt7 stems primarily from the much slower membrane dissociation rate of Syt7 ($k_{out} \sim 11\text{-}20 \text{ s}^{-1}$) compared to Syt1 ($k_{out} \sim 380\text{-}670 \text{ s}^{-1}$)^{10,11}. In contrast, without membranes, the Ca²⁺-binding affinities of Syt1 and Syt7 C2 domains are comparable^{2,12}.

To understand the Ca²⁺ activation patterns of Syt1 and Syt7 under our experimental conditions, we modeled the interaction of synaptotagmins' C2 domains with Ca²⁺ and membranes using the sequential Ca²⁺ binding and membrane loop insertion scheme depicted in Supplementary Figure 5A. As detailed in the Methods section, we constrained the model parameters as follows: k_{on} was limited by diffusion to $1 \mu\text{M}^{-1} \text{ ms}^{-1}$ and $k_{off} = 150 \text{ ms}^{-1}$ based on the intrinsic Ca²⁺ affinity $K_d = 150 \mu\text{M}$, which is similar for both Syt1 and Syt7 C2 domains^{2,12,13}. $k_{in} = 100 \text{ ms}^{-1}$ based on the characteristic time for Synaptotagmin C2 domain rotation and membrane insertion¹⁴. $k_{out} = 0.67 \text{ ms}^{-1}$ for Syt1 and $k_{out} = 0.02 \text{ ms}^{-1}$ for Syt7, determined from the apparent rates of C2 domain dissociation from lipid membranes measured in the presence of EGTA using stopped-flow experiments^{10,11,13}. Additional information and justification can be found in our recent publication¹⁵.

Indeed, the model predicts that the K_d for the Ca²⁺-dependent membrane insertion of the Syt1 and Syt7 C2 domains under steady-state conditions is $\sim 13 \mu\text{M}$ and $\sim 3 \mu\text{M}$ respectively (Supplementary Figure 5B), which is in line with the available experimental data. However, when exposed to a ramped Ca²⁺ increase to a saturating concentration of $100 \mu\text{M}$ (i.e. our experimental condition), Syt1 and Syt7 exhibit similar activation patterns for Ca²⁺-dependent membrane

insertion (Supplementary Figure 5C). Interestingly, the model suggests that under our experimental conditions, Syt7 is more sensitive to Ca^{2+} signals than Syt1. This finding contrasts with the slower kinetics of Syt7-mediated vesicular fusion, suggesting that Ca^{2+} -triggered membrane insertion of Syt7 is not the rate-limiting step in the removal of the Syt7 fusion clamp.

Supplementary Figure 5. Modeling of Ca^{2+} -dependent membrane insertion of Syt1 and Syt7 C2 domains under steady-state and experimental conditions. (A) Kinetic reaction scheme describing the Ca^{2+} -triggered membrane loop insertion of Syt1 and Syt7 C2 domains. The model parameters are outlined in the Methods section (B) Consistent with available biochemical and biophysical data^{1,2}, model-predicted Ca^{2+} dependency for C2 domain membrane loop insertion under steady-state conditions shows a higher apparent affinity for Syt7 ($K_d \sim 3 \mu\text{M}$) and a lower apparent affinity for Syt1 ($K_d \sim 13 \mu\text{M}$). (C) Under conditions of a ramped Ca^{2+} increase to a saturating concentration of 100 μM , as in our experimental setup, Syt1 and Syt7 display comparable activation patterns for Ca^{2+} -dependent membrane insertion, with Syt7 being only slightly more sensitive than Syt1

We agree that, without a detailed explanation of these considerations, our original statement in the manuscript regarding the similar activation of Syt1 and Syt7 under our experimental conditions was difficult to follow. Therefore, we have included the modeling analysis for different Ca^{2+} signals as Supplementary Figure 5 and revised the relevant results section as follows: *"Under our experimental conditions, Syt1 and Syt7 are predicated to exhibit comparable Ca^{2+} -activation patterns, with Syt7 being slightly more sensitive than Syt1 (Supplementary Figure 5)."*

In Fig 5, the revised title is still misleading. To this reviewer, this figure shows basal Ca^{2+} increases the Ca^{2+} synchronized fusion with the presence of both Syt7 and Syt1.

We respectfully disagree, as several lines of evidence that support our conclusion: (i) The enhancement on fast release component with elevated $[\text{Ca}^{2+}]_{\text{basal}}$ is observed only in the presence of Syt7. (ii) Syt1^{WT} alone does not enhance the fast component when preactivated with $[\text{Ca}^{2+}]_{\text{basal}}$. (iii) Facilitation is evident in Syt7/Syt1^{DA} conditions, where Ca^{2+} -evoked fusion is primarily mediated by Syt7 (see Figure 3B).

In the abstract, the conclusions regarding the findings in the study appear to be overstated, e.g. "...a direct demonstration that a small set of proteins is sufficient to account for how nerve terminals adapt and regulate the Ca^{2+} -evoked neurotransmitter exocytosis process...". As discussed in detail in the prior review, the findings in the reconstituted system used throughout this study are sometimes at odds with the apparent roles of these proteins in synapses. This is not particularly surprising, as reconstituted systems and a ground-up approach contains only a subset of proteins. What is needed here is to be open and transparent when the reconstituted systems fail to recapitulate the biology. The approaches used in this study are innovative and powerful, but all reconstitution studies have limitations and it incumbent on the authors to acknowledge these. The author do acknowledge, in their rebuttal, the points raised: neither

complexin nor Syt7 are physiologically relevant fusion clamps in most mammalian synapses (also indicated in point (a) above).

The concluding statement in the abstract aligns with our demonstration that a small set of proteins can account for the diverse kinetics and dynamics of Ca^{2+} -evoked neurotransmitter release observed in neuronal synapses. As recommended, we have emphasized the discrepancies between our *in vitro* findings and physiological studies in the Discussion section. We believe this approach offers a balanced perspective on the field while underscoring the significance of our results.

Since the Syt7 chimera in this manuscript contains two TMDs, is there any supplemental data showing the Syt7 chimera works similarly to WT Syt7? The authors still refer to this chimeric Syt7 as WT – this should be corrected as it is not the wild type protein.

Our initial experiments were conducted with the native Syt7^{WT} containing a single TMD. However, this construct posed technical challenges due to its low and highly variable membrane reconstitution efficiency. To address this, we modified the construct by adding a second Syt1 TMD to improve the membrane reconstitution efficiency. Preliminary control experiments show that Syt7, whether containing one or two TMDs at a 1:200 protein-to-lipid ratio, introduces a similar delay in the Ca^{2+} -evoked fusion of Syt1/VAMP2 vesicles without altering overall fusion competence.

Supplementary Figure 1C. *In vitro* functional analysis showed that the effects of the native Syt7 (Syt7^{singleTMD}) and the Syt7 construct used in this study (Syt7^{doubleTMD}) on Ca^{2+} -evoked fusion are indistinguishable.

We have revised the Materials section to include the following clarification and include the control experiment as Supplementary Figure 1C: *Our initial experiments were conducted with the full-length Syt7 wild-type protein (His⁶-SUMO-rat Synaptotagmin-7, residues 17-403). However, this construct posed technical challenges due to its low and highly variable membrane reconstitution efficiency. Hence, we modified the construct by adding a second transmembrane domain (TMD) from Syt1 with a flexible 16 residue GSGS linker, resulting in His⁶-SUMO-Syt1^{TMD}-Syt7 construct (referred to as Syt7^{WT} in this manuscript). The inclusion of Syt1^{TMD} (in addition to the Syt7^{TMD}) improved the reconstitution efficiency of the Syt7^{WT} protein into the membrane, while the flexible GSGS linker ensured the proper orientation of the two TMDs and Syt7 C2AB domains*

(Supplementary Figure 1B). Control experiments showed that effect of Syt7, whether containing one or two TMDs, on Ca²⁺-evoked fusion of Syt1/VAMP2 vesicles were indistinguishable (Supplementary Figure 1C).

REFERENCES

- 1 Sugita, S., Shin, O. H., Han, W., Lao, Y. & Sudhof, T. C. Synaptotagmins form a hierarchy of exocytotic Ca(2+) sensors with distinct Ca(2+) affinities. *EMBO J* **21**, 270-280, doi:10.1093/emboj/21.3.270 (2002).
- 2 Voleti, R., Tomchick, D. R., Sudhof, T. C. & Rizo, J. Exceptionally tight membrane-binding may explain the key role of the synaptotagmin-7 C2A domain in asynchronous neurotransmitter release. *Proc Natl Acad Sci U S A* **114**, E8518-8527, doi:10.1073/pnas.1710708114 (2017).
- 3 Ramakrishnan, S., Bera, M., Coleman, J., Rothman, J. E. & Krishnakumar, S. S. Synergistic roles of Synaptotagmin-1 and complexin in calcium-regulated neuronal exocytosis. *Elife* **9**, doi:10.7554/eLife.54506 (2020).
- 4 Huntwork, S. & Littleton, J. T. A complexin fusion clamp regulates spontaneous neurotransmitter release and synaptic growth. *Nat Neurosci* **10**, 1235-1237, doi:10.1038/nn1980 (2007).
- 5 Martin, J. A., Hu, Z., Fenz, K. M., Fernandez, J. & Dittman, J. S. Complexin has opposite effects on two modes of synaptic vesicle fusion. *Curr Biol* **21**, 97-105, doi:10.1016/j.cub.2010.12.014 (2011).
- 6 Lopez-Murcia, F. J., Reim, K., Jahn, O., Taschenberger, H. & Brose, N. Acute Complexin Knockout Abates Spontaneous and Evoked Transmitter Release. *Cell Rep* **26**, 2521-2530 e2525, doi:10.1016/j.celrep.2019.02.030 (2019).
- 7 Bera, M., Ramakrishnan, S., Coleman, J., Krishnakumar, S. S. & Rothman, J. E. Molecular Determinants of Complexin Clamping and Activation Function. *Elife* **e71938** (2022).
- 8 Malsam, J. *et al.* Complexin Suppresses Spontaneous Exocytosis by Capturing the Membrane-Proximal Regions of VAMP2 and SNAP25. *Cell Rep* **32**, 107926, doi:10.1016/j.celrep.2020.107926 (2020).
- 9 Bacaj, T. *et al.* Synaptotagmin-1 and synaptotagmin-7 trigger synchronous and asynchronous phases of neurotransmitter release. *Neuron* **80**, 947-959, doi:10.1016/j.neuron.2013.10.026 (2013).
- 10 Hui, E. *et al.* Three distinct kinetic groupings of the synaptotagmin family: candidate sensors for rapid and delayed exocytosis. *Proc Natl Acad Sci U S A* **102**, 5210-5214, doi:10.1073/pnas.0500941102 (2005).
- 11 Brandt, D. S., Coffman, M. D., Falke, J. J. & Knight, J. D. Hydrophobic contributions to the membrane docking of synaptotagmin 7 C2A domain: mechanistic contrast between isoforms 1 and 7. *Biochemistry* **51**, 7654-7664, doi:10.1021/bi3007115 (2012).
- 12 Radhakrishnan, A., Stein, A., Jahn, R. & Fasshauer, D. The Ca²⁺ affinity of synaptotagmin 1 is markedly increased by a specific interaction of its C2B domain with phosphatidylinositol 4,5-bisphosphate. *J Biol Chem* **284**, 25749-25760, doi:10.1074/jbc.M109.042499 (2009).
- 13 Davis, A. F. *et al.* Kinetics of synaptotagmin responses to Ca²⁺ and assembly with the core SNARE complex onto membranes. *Neuron* **24**, 363-376, doi:10.1016/s0896-6273(00)80850-8 (1999).
- 14 Rothman, J. E., Krishnakumar, S. S., Grushin, K. & Pincet, F. Hypothesis - buttressed rings assemble, clamp, and release SNAREpins for synaptic transmission. *FEBS Lett* **591**, 3459-3480, doi:10.1002/1873-3468.12874 (2017).

- 15 Norman, C. A., Krishnakumar, S. S., Timofeeva, Y. & Volynski, K. E. The release of inhibition model reproduces kinetics and plasticity of neurotransmitter release in central synapses. *Commun Biol* **6**, 1091, doi:10.1038/s42003-023-05445-2 (2023).

Reviewer #1

No comments to address

Reviewer #3

1) The authors have now distinguished between the biophysical properties of syt1 and syt7. However, the explanation of membrane dissociation rates of syt1 and syt7 has a discrepancy between the stated facts and k_{out} values. Specifically, syt7 has a slower membrane dissociation rate ($k_{diss} \sim 19.7 \text{ s}^{-1}$) than the faster rate of syt1 ($k_{diss} \sim 378 \text{ s}^{-1}$). In modeling membrane insertion of the loops of syts (Supplementary Fig. 5), the authors have used incorrect values of k_{out} for syt1 and syt7; in fact, the values they used are almost interchanged between the two isoforms (with syt1 being slow and syt7 being fast).

We have checked both the manuscript and Supplementary Figure 5 and did not find any discrepancies between the data and the model parameters used. The constraining of k_{out} values for Syt1 and Syt7 has been described in detail in our previous work (Norman et al 2023) as follows:

“The dissociation kinetics of Syt1 and Syt7 C2 domains from the membranes have previously been determined in stopped flow experiments where Ca^{2+} is rapidly removed from the system after fast dilution in an EGTA-containing buffer. The membrane dissociation curves closely follow single exponential decay functions with dissociation rate constants (k_{diss}) in the range of $0.38 - 0.7 \text{ ms}^{-1}$ for Syt1 and $0.008 - 0.02 \text{ ms}^{-1}$ for Syt7⁹⁻¹¹. In this work we used representative values of $k_{diss} = 0.5 \text{ ms}^{-1}$ for Syt1 and $k_{diss} = 0.015 \text{ ms}^{-1}$ for Syt7.

Modelling the stopped flow experiment, the kinetic scheme of Ca^{2+} binding and membrane insertion of synaptotagmin C2 domains shown in Figure 1B can be reduced to

in the kinetic model in Figure 1B (states S_0 and S_1), can be combined into a single absorbing state S_{0-1} . The dynamics of the non-absorbing states, S_2 and I , are described by the system of first order ordinary differential equations:

$$\frac{dP(S_2)}{dt} = k_{out}P(I) - (k_{in} + 2k_{off})P(S_2),$$

$$\frac{dP(I)}{dt} = k_{in}P(S_2) - k_{out}P(I),$$

where $P(A)$ indicates the probability that the system occupies state A at time t . The general solution for this system can be expressed as

$P(I) = Ae^{\lambda_1 t} + Be^{\lambda_2 t}$, where A and B are constants and the rate parameters are given by

$$\lambda_1 = \frac{-(k_{out} + 2k_{off} + k_{in}) + \sqrt{(k_{out} + 2k_{off} + k_{in})^2 - 8k_{off}k_{out}}}{2},$$

$$\lambda_2 = \frac{-(k_{out} + 2k_{off} + k_{in}) - \sqrt{(k_{out} + 2k_{off} + k_{in})^2 - 8k_{off}k_{out}}}{2}.$$

From the solutions for λ_1 and λ_2 , using the constrained values of $k_{on} = 1 \mu\text{M}^{-1} \text{ms}^{-1}$, $k_{off} = 150 \text{ms}^{-1}$, and $k_{in} = 100 \text{ms}^{-1}$, the fast component λ_2 has a magnitude greater than 400ms^{-1} for all positive values of k_{out} , and would therefore dissipate well within the dead time of the stopped-flow apparatus ($> 1 \text{ms}$)¹⁰. This means that the slower exponential component dominates the model dynamics over timescales observed in the stopped-flow experiments, and the simplification $P(I) \propto e^{\lambda_1 t}$ should provide an appropriate approximation of the experimental data. The full expansion of λ_1 can then be equated with the apparent membrane dissociation rate of Syt1 or Syt7, k_{diss} , to complete the system of kinetic

parameters with: $k_{out} = k_{diss} \left(1 - \frac{k_{in}}{k_{diss} - 2k_{off}} \right)$. For the value of $k_{diss} = 0.5 \text{ms}^{-1}$ this equation yields $k_{out} = 0.67 \text{ms}^{-1}$ for Syt1, and for $k_{diss} = 0.015 \text{ms}^{-1}$ it yields $k_{out} = 0.02 \text{ms}^{-1}$ for Syt7.

We have revised the method section to include reference to our previous work as follows:

$k_{out} = 0.67 \text{ms}^{-1}$ for Syt1 and $k_{out} = 0.02 \text{ms}^{-1}$ for Syt7 was determined from the apparent rates of C2 domain dissociation from lipid membranes (k_{diss}) measured in the presence of EGTA using stopped-flow experiments⁵⁷⁻⁵⁹ as described in our previous work¹².

We note that, while a higher K_{out} value results in a slower membrane dissociation rate for Syt7 compared to Syt1 when Ca^{2+} is removed from the system, the elevated K_{out} also leads to more efficient membrane insertion of Syt7 during ramped increase in $[\text{Ca}^{2+}]$. This is because the higher K_{out} enhances the apparent Ca^{2+} /membrane affinity of Syt7. Indeed, this is reflected in model output in Supplementary Figure 5C.

2) 2) As suggested by Reviewer #1, 'facilitation' should not be used to describe in vitro findings (see Fig. 5 legend).

We have revised Figure 5 legend to read 'enhancement of Ca^{2+} -synchronized release' instead of 'facilitation'.